# Characterization of the Src-regulated kinome identifies SGK1 as a key mediator of Src-induced transformation

Xiuquan Ma [1,2], Luxi Zhang[1,2], Jiangning Song [2,3,4], Elizabeth Nguyen[1,2], Rachel S. Lee[1,2], Samuel J. Rodgers[1,2], Fuyi Li[2,3], Cheng Huang[5], Ralf B. Schittenhelm[5], Howard Chan[1,2], Chanly Chheang[1,2], Jianmin Wu [6], Kristin K. Brown[7,8,9], Christina A. Mitchell[1,2], Kaylene J. Simpson[9,10] & Roger J. Daly[1,2]

Despite significant progress, our understanding of how specific oncogenes transform cells is still limited and likely underestimates the complexity of downstream signalling events. To address this gap, we use mass spectrometry-based chemical proteomics to characterize the global impact of an oncogene on the expressed kinome, and then functionally annotate the regulated kinases. As an example, we identify 63 protein kinases exhibiting altered expression and/or phosphorylation in Src-transformed mammary epithelial cells. An integrated siRNA screen identifies nine kinases, including SGK1, as being essential for Src-induced transformation. Accordingly, we find that Src positively regulates SGK1 expression in triple negative breast cancer cells, which exhibit a prominent signalling network governed by Src family kinases. Furthermore, combined inhibition of Src and SGK1 reduces colony formation and xenograft growth more effectively than either treatment alone. Therefore, this approach not only provides mechanistic insights into oncogenic transformation but also aids the design of improved therapeutic strategies.

[1] Cancer Program, Biomedicine Discovery Institute, Monash University, Melbourne, VIC 3800, Australia. [2] Department of Biochemistry and Molecular Biology, Monash University, Melbourne, VIC 3800, Australia. [3] Infection and Immunity Program, Biomedicine Discovery Institute, Monash University, Melbourne, VIC 3800, Australia. [4] Monash Centre for Data Science, Faculty of Information Technology, Monash University, Melbourne, VIC 3800, Australia. [5] Monash Biomedical Proteomics Facility and Monash Biomedicine Discovery Institute, Monash University, Melbourne, VIC 3800, Australia. [6] Key Laboratory of Carcinogenesis and Translational Research (Ministry of Education/Beijing), Centre for Cancer Bioinformatics, Peking University Cancer Hospital & Institute, Beijing 100142, China. [7] Cancer Therapeutics Program and Cancer Metabolism Program, Peter MacCallum Cancer Centre, Melbourne, VIC 3000, Australia. [8] Department of Biochemistry and Molecular Biology, The University of Melbourne, Melbourne, VIC 3010, Australia. [9] Sir Peter MacCallum Department of Oncology, The University of Melbourne, Melbourne, VIC 3010, Australia. [10] Victorian Centre for Functional Genomics, Peter MacCallum Cancer Centre, Melbourne, VIC 3000, Australia. These authors contributed equally: Xiuquan Ma, Luxi Zhang. Correspondence and requests for materials should be addressed to R.J.D. (email: roger.daly@monash.edu)

While great progress has been made in characterizing downstream signaling mechanisms of specific tyrosine kinase oncogenes, most of this work has focused on well-established signaling pathways, such as the Ras/MAPK, PI3K/Akt, and JAK/Stat pathways[1]. This continues despite data from cancer genome sequencing analyses, mass spectrometry (MS)-based proteomics and functional genomic screens highlighting involvement of many poorly-characterized protein kinases in cell transformation[2]. Consequently, our understanding of oncogenic kinase signaling is clearly limited and likely underestimates the complexity of downstream signaling events and their functional roles.

Src was the first cellular proto-oncogene to be identified[3] and is negatively regulated by phosphorylation on a conserved C-terminal tyrosine residue (Y527 and Y530 in chicken and human Src, respectively), mediated by C-terminal Src kinase (Csk). This promotes formation of a closed, inactive conformation where the phosphorylated tyrosine residue is engaged by the src homology (SH)2 domain. Reflecting this, the Src Y527F mutant is constitutively active and exhibits transforming activity[4]. While Src mutations in human cancers are rare, increased Src expression and activity occurs in a variety of malignancies, including breast, non-small cell lung, colon, and pancreatic cancers, where it correlates with poor prognosis or mediates resistance to specific therapies[5–9]. Reflecting this, several Src-directed targeted therapies are currently in clinical trials in solid malignancies, including the tyrosine kinase inhibitors saracatanib, bosutinib, and dasatinib. However, disease response or stabilization following treatment with Src Tyrosine Kinase Inhibitors (TKIs) has been generally limited to small subsets of patients[10], highlighting the need for a greater understanding of Src-induced transformation and identification of biomarkers that predict patient response to such therapies.

Src signaling regulates a variety of biological endpoints, including cell proliferation, survival, adhesion, migration, and invasion[11,12], and several approaches have been utilized to interrogate substrates, signaling pathways and transcriptional programs regulated by this oncogene. Early work exploited monoclonal antibody generation and/or expression cloning approaches to identify Src substrates[13,14], while transcript profiling has identified gene expression programs associated with cell cycle control, cytoskeletal organization, cell adhesion, and motility as being regulated by Src[15–17]. Importantly, this work has been complemented and greatly extended by the application of an immunoaffinity-coupled MS-based proteomics workflow, where tryptic tyrosine-phosphorylated peptides are enriched prior to MS analysis[18]. Application of this approach to Src-transformed fibroblasts and cancer cells exhibiting high levels of Src activity has highlighted the diversity of protein classes that are tyrosine-phosphorylated upon Src-induced transformation, ranging from specific kinases and phosphatases to GEFs, GAPs, and scaffolds, and revealed novel processes regulated by Src such as RNA maturation[19–23].

Despite these advances in our understanding of Src-induced oncogenesis, the protein kinase pathways and networks that regulate the pleiotropic effects of active Src remain poorly characterized, since the proteomic approaches applied so far have focused on the tyrosine phosphoproteome, and do not provide insights into the expression or activation status of the large numbers of non-tyrosine phosphorylated kinases that lie downstream. This is a critical issue, given the biological significance and translational relevance of this oncogene.

To address this, we here apply a quantitative, MS-based kinome profiling workflow that determines the expression and activation status of the majority of the expressed kinome in Src-transformed cells[24–27]. In addition, and in contrast to the majority of related profiling studies, we integrate this approach with a functional genomic screen that annotates the identified Src-regulated kinome, thereby identifying kinases that are essential for, or modulate, Src-induced transformation. This approach not only provides major mechanistic insights into Src-mediated transformation, but may also aid the design of improved therapeutic strategies for cancers expressing active Src.

## Results

**Identification of the Src-regulated kinome.** We selected MCF-10A immortalized mammary epithelial cells expressing a constitutively active version of Src (Src Y527F) as an appropriate system for characterization of the Src-regulated kinome, as these cells exhibit a transformed phenotype in culture and model the Src family kinase signaling network characteristic of basal/triple negative breast cancer (TNBC) cells[20]. Control MCF-10A (MCF-10A_Ctrl) cells and their Src-transformed counterparts (MCF-10A_Src) were differentially SILAC labeled and cell lysates subjected to a large-scale kinome purification workflow followed by kinase detection and purification by LC-MS/MS (Fig. 1a)[27]. Amongst the detected protein kinases, 59 and 6 exhibited consistent site-selective phosphorylation or expression changes of ≥1.5-fold, respectively (see Methods, Supplementary Data 1–2 and Supplementary Table 1). Src activation impacted representatives of all of the major kinase families (Fig. 1b, c), but approximately two-thirds of the increased phosphosites corresponded to tyrosine kinases, with the next largest contribution being derived from the CAMK family (Fig. 1c). For downregulated sites, the tyrosine kinase, CAMK, and AGC families made the largest and similar contributions (Fig. 1c). Certain kinases exhibited complex phosphorylation changes. For example, the non-receptor tyrosine kinase FAK displayed increased phosphorylation on multiple sites encompassing serine, threonine and tyrosine-residues, and decreased phosphorylation on three serines (Fig. 1d). The receptor tyrosine kinase (RTK) EPHA2 also exhibited multisite phosphorylation changes, including enhanced phosphorylation of a triplet of threonine residues within the activation loop (Fig. 1d).

Activated Src would be expected to initiate a cascade of phosphorylation events, starting with direct phosphorylation by Src itself, and then those mediated by downstream kinase pathways, often involving serine/threonine kinases. Tyrosine phosphorylation constituted 36.3% of altered kinase phosphorylation sites (Supplementary Data 1). Characterization of this suite of altered phosphosites by pLogo determined that upregulated tyrosine phosphorylation sites were enriched for acidic (D/E) residues at −3 and −4 relative to the tyrosine (Supplementary Figure 1a), consistent with the consensus motif for Src itself[28]. Phosphorylation events on kinases mediated by basophilic kinases and proline-directed kinases constituted approximately one-quarter of all altered phosphorylation sites (Supplementary Figure 1b). Upregulated serine phosphorylation sites exhibited enrichment for phosphorylation by basophilic kinases, and a lesser preference for phosphorylation by proline-directed kinases (Supplementary Figure 1a). Upregulated threonine phosphorylation sites did not exhibit the same enrichment for phosphorylation by basophilic kinases as serine sites (Supplementary Figure 1a).

Visualization of interactomes amongst the regulated kinases revealed that Src represented the focal point of a dense network containing regulators of cytoskeletal organization that included the tyrosine kinases EPHA2, EPHA5, EPHB3, EPHB4, ABL1/2, and PTK2 (FAK), and the low-molecular-weight G proteins RHOA and CDC42 (Supplementary Figure 2a-b). A further prominent feature of this network was the presence of many proteins involved in the PI3K signaling pathway (PIK3CA, PIK3CB, MTOR, AKT2, SGK1, PHLPP1/2). Other interaction

hubs included ones centered on regulators of protein folding (HSP90AA1) and cell cycle control (CDK6, TP53).

Assignment of regulated kinases to known cellular pathways and processes highlighted the impact of Src on a variety of receptor and non-receptor tyrosine kinases (Fig. 2). Src

transformation was associated with decreased expression of EPHB3, and in addition to the complex phosphorylation changes on FAK and EPHA2 mentioned previously, increased phosphorylation of several RTKs including EGFR, DDR1, and AXL. Expression of oncogenic Src also led to increased phosphorylation

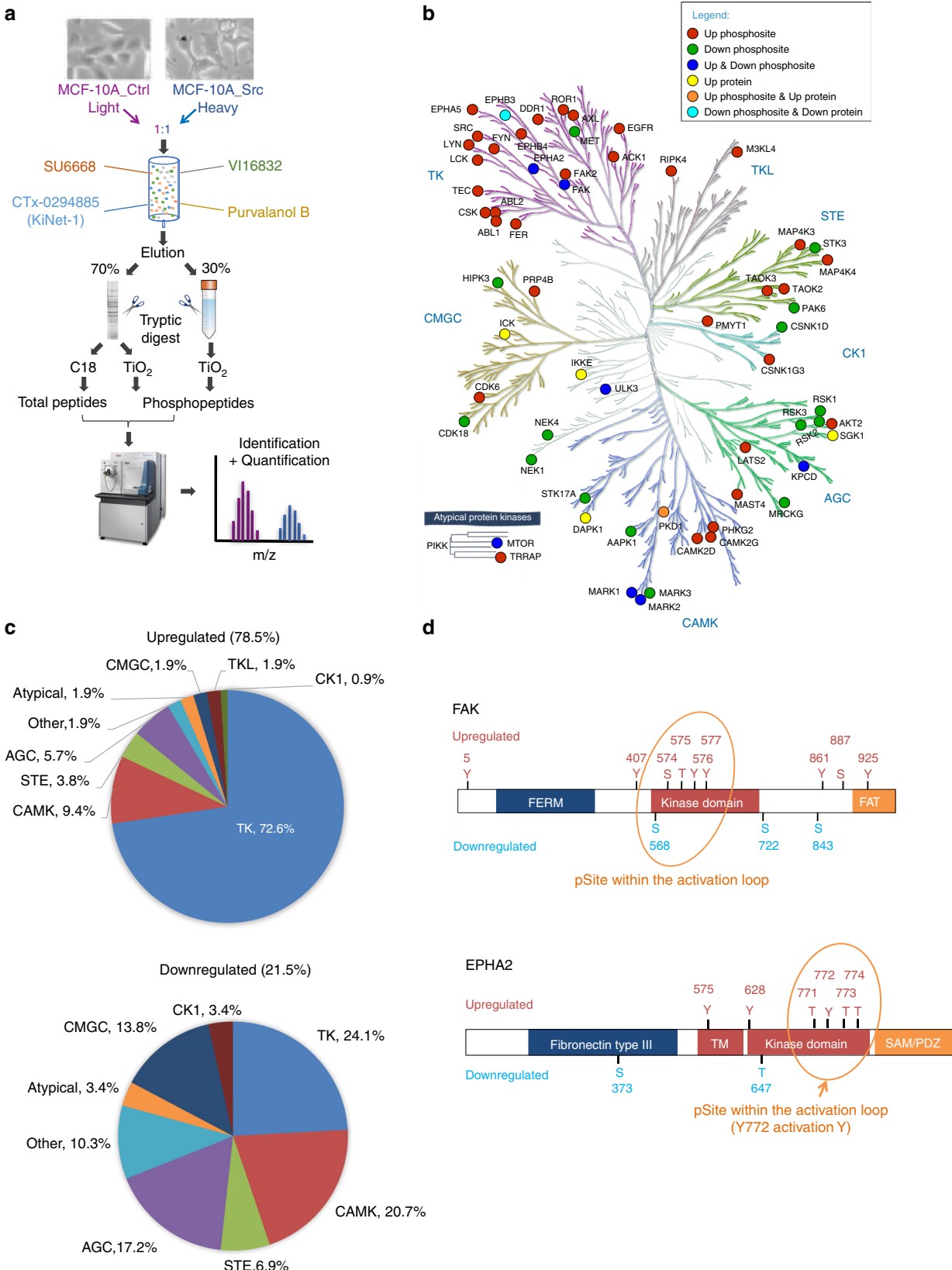

of multiple non-receptor tyrosine kinases, including CSK and LCK and two without previously identified functional links to Src in FER and TEC. Consistent with previous data from our laboratory[29], expression of active Src in this system enhanced PI3K pathway activation (as evidenced by increased phosphorylation of AKT2 on S474 and SGK1 expression) but decreased activation of the ERK MAPK pathway (as indicated by reduced phosphorylation of RSKs). Src-induced transformation also perturbed phosphorylation of many kinases with reported upstream roles in the JNK and p38 MAPK pathways, including MAP4K3-4 and TAOK2-3.

Our data further emphasized the impact of oncogenic Src on diverse cellular processes. While Src is a known regulator of cytoskeletal organization, our work significantly extends our knowledge of downstream kinase mediators to include three members of the MARK family[30], and the atypical CDK, CDK18[31]. The Src-regulated kinome also included kinases implicated in regulation of cell metabolism (e.g. PHKG2[32]), ciliogenesis (ICK[33]), mRNA splicing (PRP4B[34]), DNA repair, and the cell cycle (specific NEK family members[35]).

One of the reasons for selecting the MCF-10A_Src system for this study was that it provides an excellent model for TNBC cells, where Src family kinases play a prominent signaling role[20]. Indeed, MCF-10A_Src and TNBC cells share similar signaling network characteristics, and many proteins exhibiting enhanced tyrosine phosphorylation in the MCF-10A_Src model are also highly phosphorylated in TNBC cells, where treatment with selective Src inhibitors reduces their tyrosine phosphorylation levels[20]. In order to determine the extent to which our Src-regulated kinome was under Src regulation in TNBC, we undertook small scale kinome profiling on MDA-MB-231 TNBC cells treated with the small molecule Src inhibitor AZD0530. We accept that there are problems with this approach, which include the marked difference in genetic background between the two cell types and contrasting culture conditions. However, while we did not attain the degree of coverage that characterized the original profiling of MCF-10A_Src cells, for sites that overlapped between the two datasets and exhibited enhanced phosphorylation in MCF-10A_Src cells, the phosphorylation of >80% of these were decreased by AZD0530 treatment of MDA-MB-231s, i.e., the sites were regulated in the same direction (Supplementary Table 2). While the regulated kinases included EPHA2, LYN, ABL2 and FYN, which are also bound by AZD0530 and hence may reflect off-target effects[36], they also included kinases not known to be targeted by AZD0530 and identified by our profiling of MCF-10A_Src cells, including TEC, MARK2, and NEK1. These data support the validity of the MCF-10A_Src model and our approach.

**Functional characterization of Src-regulated kinases.** While we could undertake a preliminary assignment of the majority of Src-regulated kinases to functional categories based on previously published work, many of them are relatively understudied, with unknown roles in Src-induced transformation. This highlighted

the need for further functional interrogation of the Src-regulated kinome. To this end, we first determined phenotypic differences between MCF-10A_Ctrl and MCF-10A_Src cells. This revealed that the two cell types proliferated at similar rates in full medium containing EGF (referred to as +EGF), but Src-transformed cells exhibited greater proliferation in medium containing low serum that lacked EGF (referred to as −EGF) (Fig. 3a). Moreover, in 3D Matrigel culture, MCF-10A_Ctrl cells were unable to form acini in −EGF medium, whereas Src-transformed cells formed large, aberrant acini (Fig. 3b). The latter observation was consistent with our previous report that expression of active Src in this system overcomes growth factor dependency and disrupts the normal morphogenetic program, resulting in spheroids with filled lumens (due to impaired apoptosis) and disrupted polarity[29]. Having established these parameters, we set out to characterize the dependency of the two cell types on kinases identified by our kinomic screen, as well as on other proteins identified by ourselves[20] or others[23] to exhibit altered phosphorylation in Src-transformed cells. This was undertaken by screening with a custom siRNA SMARTpool library encompassing 183 targets (Supplementary Data 3). Both cell types were utilized in 2D screens with cell viability as an endpoint (Supplementary Data 4), while Src-transformed cells were also screened under 3D conditions, −EGF, with acinar size as an endpoint (Supplementary Data 5).

In general, under 2D conditions Src-transformed cells were significantly less sensitive to knockdown of library targets than control cells (Fig. 3c and Supplementary Figure 3a-b). Comparison of how knockdown of individual kinases impacted viability of control and Src-transformed cells under −EGF conditions identified kinases that were markedly required for viability of both cell types (Supplementary Figure 3c), or only control cells (Fig. 3d). The former included components of key proliferative pathways such as MAPK1, whereas the latter included EGFR, consistent with the role of Src family kinases (SFKs) in promoting resistance to EGFR-directed therapies[7]. Validation of knockdown for specific kinases is shown in Supplementary Figure 3d. Interestingly, whereas knockdown of numerous kinases preferentially impacted control cells (Fig. 3d), there were no targets that exhibited a selective effect on Src-transformed cells. The dependency on certain kinases varied according to cell type and culture condition. For example, knockdown of MAP4K5 exerted contrasting effects in the two cell types in the absence of EGF (Fig. 3e).

The screen was also optimized and undertaken under 3D conditions, −EGF, which provided a more robust assay for Src-induced transformation (Figs 3b and 4a). In the primary screen, knockdown of 61 targets decreased acinar size, while 4 enhanced it (Supplementary Data 5, Fig. 4b). In order to prioritize targets for validation, we excluded kinases where knockdown severely impacted 2D proliferation (≥30% inhibition) (Supplementary Figure 4), limiting the list to 34 targets. These targets were subjected to validation by deconvolution of the corresponding siRNA SMARTpools and testing of individual siRNAs (Fig. 4c).

**Fig. 1** Definition of the Src-regulated kinome. **a** MS-based kinome profiling workflow. Lysates from differentially SILAC-labeled control and MCF-10A_Src cells were mixed and then subjected to kinome enrichment. Following in-gel or in-solution tryptic digestion, additional purification of phosphopeptides was undertaken using TiO$_2$ beads prior to MS analysis. **b** Distribution of Src-regulated kinases over the human kinome tree. Regulation at the expression and/or phosphorylation level is indicated by corresponding shading, as defined in the legend. Eight main kinase groups are highlighted: AGC (containing protein kinases A, G, and C); CAMK (calcium/calmodulin-dependent protein kinase); CK1 (casein kinase 1); CMGC (containing cyclin-dependent kinase, MAPK, glycogen synthase kinase 3 and CDC2-like); STE (homologs of yeast sterile 7, sterile 11 and sterile 20); TK (tyrosine kinase); TKL (tyrosine kinase-like) and 'atypical' protein kinases (which includes ABC1, Alpha, Brd, PDHK, PIKK, RIO, and TIF1 kinases). **c** Contribution of individual kinase families to upregulated or downregulated kinase phosphosites. **d** Schematic representation of up- and downregulated FAK and EPHA2 phosphosites

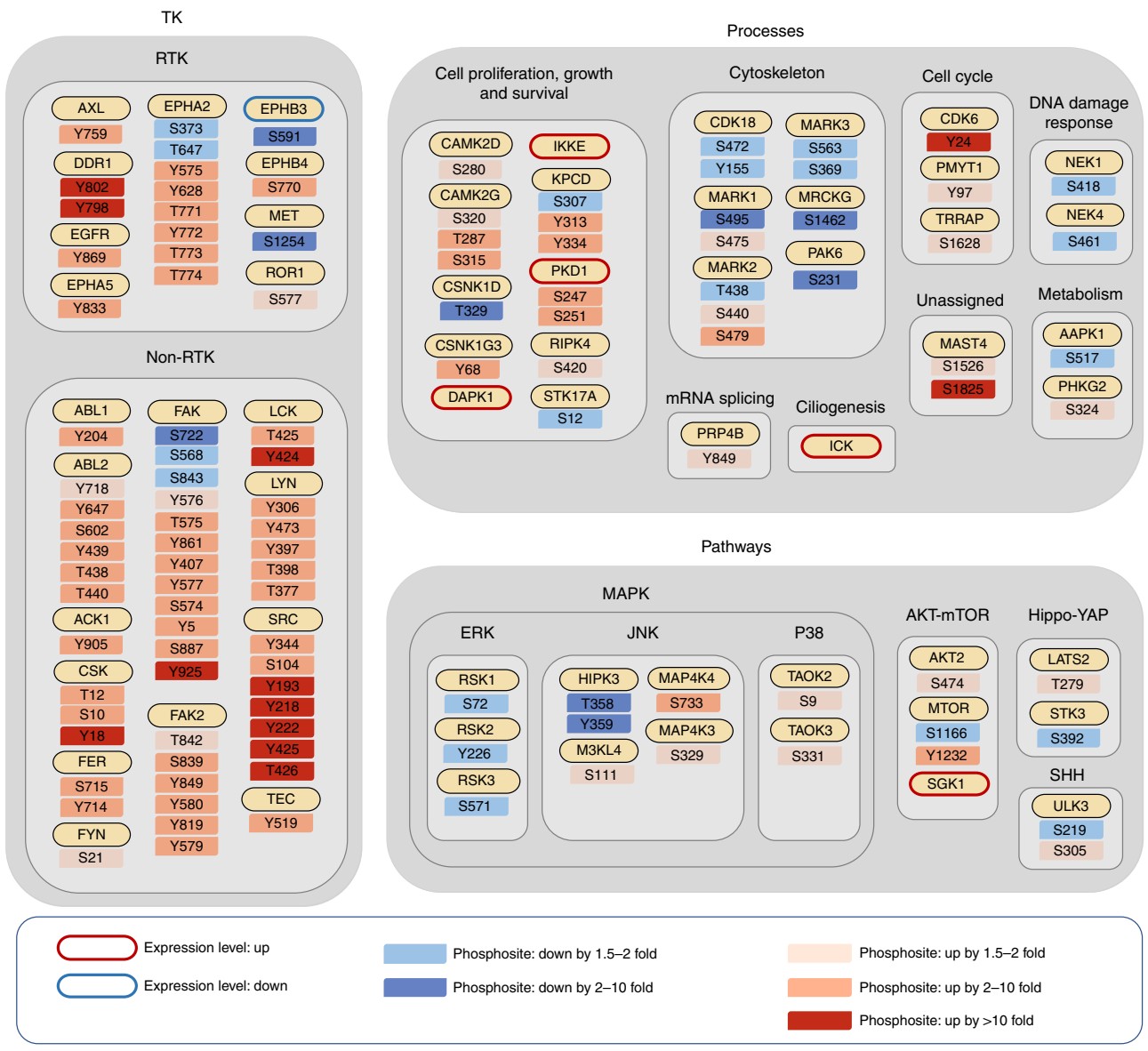

**Fig. 2** Bioinformatic assignment of Src-regulated kinases. Kinases were assigned to specific kinase classes, biological processes and pathways according to the information extracted from the GeneCards database and/or the literature. Expression and phosphosite regulation is as defined in the legend. Fold change is expressed relative to the value for vector control cells which was arbitrarily set at 1. Up or down means levels were increased or decreased in MCF-10A_Src cells compared to MCF-10A_Ctrl cells, respectively. Kinases are in alphabetical order in each group and sites for each kinase are arranged in ascending order of fold change

This identified 10 kinase targets that markedly perturbed the Src-induced transformed phenotype in 3D culture (Table 1, Supplementary Figure 5 and Supplementary Table 3). While 9/10 targets decreased acinar size upon knockdown, ablation of MAP4K5 enhanced it (Fig. 4b and Supplementary Table 3). Five of the 10 validated kinases are present in the identified Src-regulated kinome (Figs. 1–2, Supplementary Data 1). The other five kinases were included in the functional screens but did not meet the ≥1.5-fold cut-off utilized in the kinome screen. However, interrogation of the kinomic profiling data revealed that 4/5 of these do exhibit modest phosphorylation changes in Src-transformed cells (Table 1).

**SGK1 is required for Src-mediated transformation.** Serum and glucocorticoid-regulated kinase 1 (SGK1) (Fig. 5a) exhibited markedly enhanced expression in MCF-10A_Src cells and was a

validated hit in the 3D functional screen (Table 1). SGK1 is a member of a family of three serine/threonine kinases, all exhibiting significant homology to the Akt kinase family[37]. Increased SGK1 expression (the multiple SGK1 bands represent different isoforms generated by use of alternative translation start sites[37] or reflect phosphorylation of SGK1[38]), and phosphorylation of its downstream substrate N-myc downstream regulated 1 (NDRG1), was confirmed for Src-transformed cells in both full medium (+EGF) and under –EGF conditions in 2D culture (Fig. 5b). Under the latter conditions, Src-transformed cells exhibit a marked increase in Erk activation, and treatment with a selective MEK inhibitor determined that this contributes to SGK1 upregulation (Fig. 5c). However, since Src-transformed cells exhibit increased YAP transcriptional activity (Supplementary Figure 6a), and a recent study demonstrated that SGK1 represents a YAP target gene[39], we also determined the effect of treatment with the YAP inhibitor verteporfin[40] or YAP knockdown in Src-

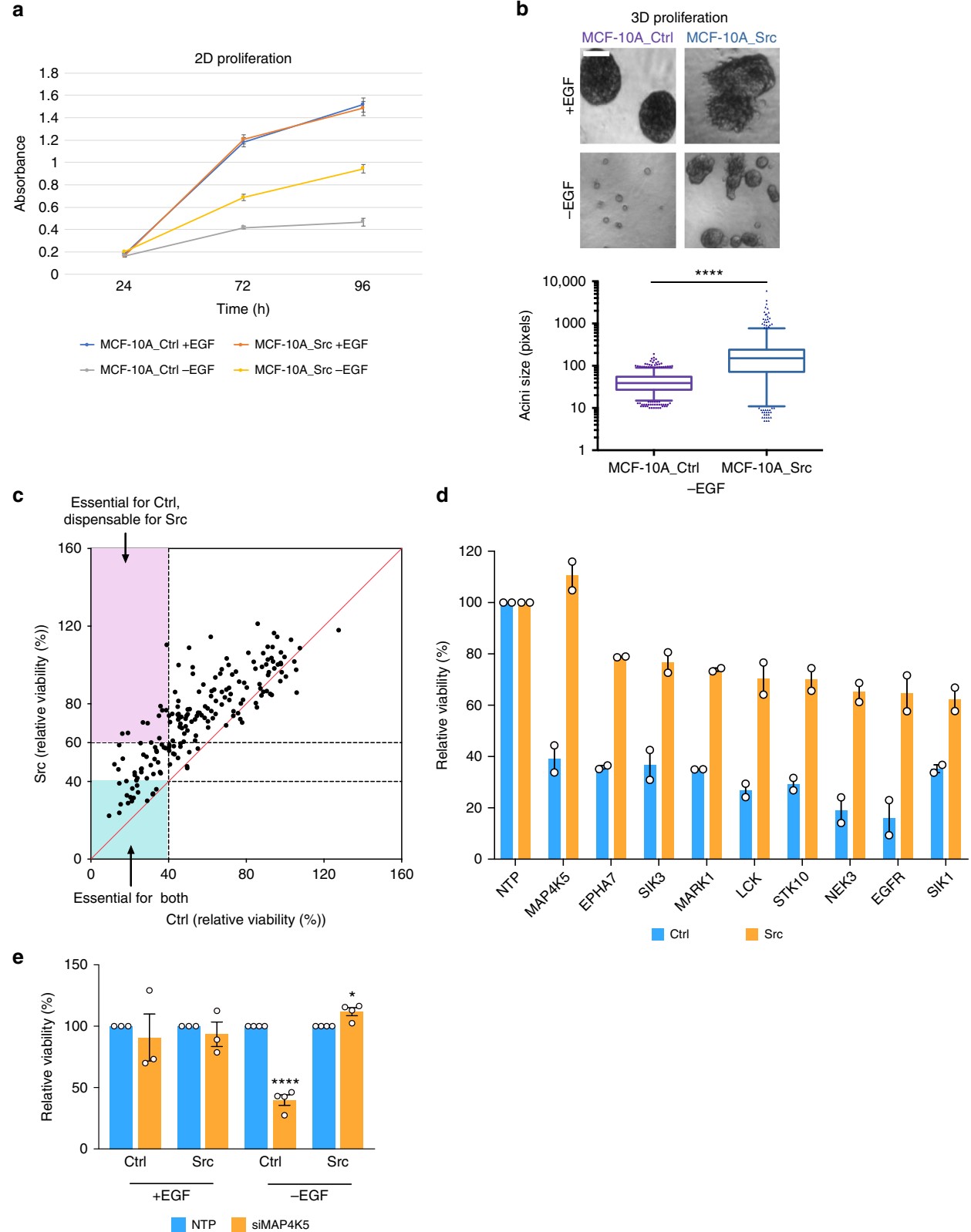

transformed cells. This resulted in decreased SGK1 levels (Fig. 5d, e), indicating that both MEK-Erk and YAP signaling promote SGK1 expression in Src-transformed cells. While knockdown of SGK1 reduced growth of MCF-10A_Src acini (Fig. 4c), SGK1 overexpression enhanced it (Fig. 5f), identifying SGK1 as an important positive regulator of Src-induced transformation. Treatment with a selective SGK1 kinase inhibitor[41]

(Supplementary Figure 6b) reduced acinar growth (Fig. 5g), providing strong evidence that this role of SGK1 is kinase-dependent.

In order to determine the biological basis for SGK1 dependency in MCF-10A_Src acini, we examined cleaved Caspase-3 and Ki67 levels by immunofluorescent microscopy as markers of apoptosis and proliferation, respectively (Fig. 6a). This

**Fig. 3** Functional annotation of Src-regulated kinases. **a** Monolayer (2D) proliferation of MCF-10A control (_Ctrl) and MCF-10A_Src cells, −/+EGF. MTS assays were undertaken in 96-well plate as described in Methods. Error bars represent s.d. from five replicates. **b** Growth characteristics of MCF-10A_Ctrl and MCF-10A_Src cells in 3D Matrigel culture, −/+EGF. Cells were plated in 3D culture −/+EGF in a 96-well plate as described in Methods. Acini were imaged (Scale bar represents 50 μm) for both +EGF and −EGF conditions (top panels) and for the latter condition, the size of >100 acini were measured and plotted on a box and whisker graph, with the center line indicating the median, the upper and lower bounds of box indicating the 25th and 75th percentiles of the data, and the upper and lower whiskers indicating the 5th and 95th percentiles of the data (bottom panel). ****$p < 0.0001$ by Mann–Whitney test. **c** Kinase-dependency of control and Src-transformed cells under −EGF conditions. siRNA SMARTpools of targets were robotically reverse transfected in duplicate 96-well plates at 40 nM and cell viability measured 96 h post-transfection. Relative viability values for MCF-10A_Ctrl (Ctrl) and MCF-10A_Src (Src) cells were plotted. Each dot indicates a kinase screened in both Ctrl and Src cells. Red line is the line of equality ($y = x$). Relative viability <40% in both Ctrl and Src cell lines is defined as "essential for both". Relative viability <40% in Ctrl cells, and >60% in Src cells is defined as "essential for Ctrl, dispensable for Src". No kinases were essential for Src cells, dispensable for Ctrl. **d** Kinases required preferentially by controls. Targets identified in (**c**) as "essential for Ctrl, dispensable for Src" were plotted. Error bars represent s.e.m. from two biological replicates. **e** Dependency on MAP4K5 is growth factor- and cell type-dependent. Viability is expressed relative to non-targeting control (NTP) which was arbitrarily set at 100%. Error bars represent s.e.m. from three or four biological replicates. *$p < 0.05$, ****$p < 0.0001$ by Student's $t$-test

revealed a significant increase in the number of acini positive for cleaved Caspase-3 upon SGK1 knockdown or SGK inhibitor treatment, while Ki67 staining was unaffected, indicating that the effect of these manipulations on acinar size reflects increased apoptosis (Fig. 6b). Depending on the cell system, SGK1 can signal to a variety of downstream effectors[37]. While SGK1 knockdown did not decrease phosphorylation of candidate effectors Erk, GSK3 and FOXO3A, or JNK and p38 MAPK (Supplementary Figure 6c), it did reduce mTOR activation, as determined by blotting for S6 kinase pS235/6 (Fig. 6c). This dependency of mTOR activation on SGK1 was greater in 3D culture, as treatment of MCF-10A_Src acini with the SGK1 inhibitor resulted in a marked diminution of S6 activation (Fig. 6d). These data are consistent with a recent report that SGK1 regulates mTOR through phosphorylation of TSC2[41], and identify a SGK1-mTOR pathway that is critical for Src transformation in this system. Further supporting the functional role of mTOR in this setting, treatment of 3D cultures with the mTOR inhibitor rapamycin almost completely abolished acini formation (Supplementary Figure 6d), and both Akt2 and mTOR were identified as hits in the 3D functional screen (Supplementary Data 5).

**SGK1 signals downstream of Src in specific human cancers.** In order to validate the oncogenic role of SGK1 downstream of Src, we first addressed the role of SGK1 in breast cancer, since Src plays key roles in this malignancy[6]. Interrogation of publicly-available gene expression data revealed that SGK1 mRNA expression is higher in the basal subtype compared to the luminal A, luminal B and HER2 ones (Supplementary Figure 7a), and consistent with these data, western blotting confirmed enhanced SGK1 protein expression in TNBC cell lines compared to their luminal counterparts (Fig. 7a). Since basal/TNBC cell lines exhibit enhanced SFK activation compared to luminal ones (Fig. 7a) and a prominent SFK signaling network[20], we characterized the impact of Src inhibition in three TNBC cell lines, Hs578T, MDA-MB-231 and -468. Treatment with the selective Src inhibitors AZD0530 or PP2 resulted in a marked reduction in SGK1 expression (Fig. 7b, c, Supplementary Figure 8), as did Src knockdown (Fig. 7d) indicating that SGK1 is under Src regulation in these cells as well as in Src-transformed MCF-10As (Fig. 5b). To determine the functional role of SGK1, we established stable pools of MDA-MB-231s expressing tetracycline-inducible shRNAs targeting SGK1 for long term SGK1 knockdown (Fig. 7e). Knockdown of SGK1 significantly reduced colony formation (Fig. 7f), as did treatment with the SGK1 inhibitor (Fig. 7g). However, since Src inhibition did not completely abolish SGK1 expression (Fig. 7b), we hypothesized that targeting of both kinases would be more effective. Indeed, combining SGK1

knockdown with AZD0530 treatment led to a greater inhibition of colony formation than either manipulation alone (Fig. 7h). These effects were also observed with a different selective Src inhibitor (PP2) and an independent SGK1 shRNA (Supplementary Figure 9a-c). In addition, the combination of small molecule SGK1 and Src inhibitors revealed a moderate synergistic effect of combined kinase targeting (combination index ~0.8) (Fig. 7i, Supplementary Figure 9d).

In order to determine whether these observations extend to a different cancer type, we utilized A549 non-small cell lung cancer (NSCLC) cells, given the known role of Src in progression of this malignancy[9]. As with TNBC cells, use of selective Src inhibitors or Src knockdown markedly reduced SGK1 expression (Fig. 8a, b, Supplementary Figure 10a). In addition, combining Src inhibitor treatment with SGK1 knockdown was more effective than the individual treatments in suppressing colony growth (Fig. 8c, d, Supplementary Figure 10b-d), and the combination of Src and SGK1 inhibitors was synergistic (Fig. 8e).

Finally, we characterized how combined Src and SGK1 targeting affected tumor growth in vivo. To do this, we exploited the MDA-MB-231 TNBC model expressing inducible SGK1 shRNA (Fig. 7e), and grew these cells as orthotopic tumors in nude mice. Doxycycline-induced knockdown of SGK1 was confirmed by western blotting (Supplementary Figure 11). Treatment with the Src inhibitor AZD0530 resulted in a modest but significant reduction in tumor growth, while the effect of SGK1 was more marked. However, combination of the two treatments resulted in a significant reduction in tumor size compared to either manipulation alone (Fig. 9a, b), validating the relevance of our in vitro experiments (Figs. 7, 8). Overall, these data provide important mechanistic insights into Src signaling in human cancer, and highlight how dissection of the Src signaling network can lead to improved therapeutic strategies.

**Discussion**

In this study, we describe an approach for interrogating the mechanism of cellular transformation by a specific oncogene, which integrates MS-based definition of global kinome perturbations with siRNA screens that functionally annotate the identified oncogene-regulated kinome. We have utilized a robust 3D growth assay under growth factor-restricted conditions as the functional read-out, but characterization could be readily extended to other biological endpoints such as cell migration. Overall, our integrated screen provides major insights into how expression of oncogenic Src impacts the expressed kinome, identifies kinases, some poorly characterized, essential for cell transformation by this oncogene or that modulate this process, and reveals a strategy to improve therapeutic targeting of Src. We accept that the approach could be further refined through use of an inducible

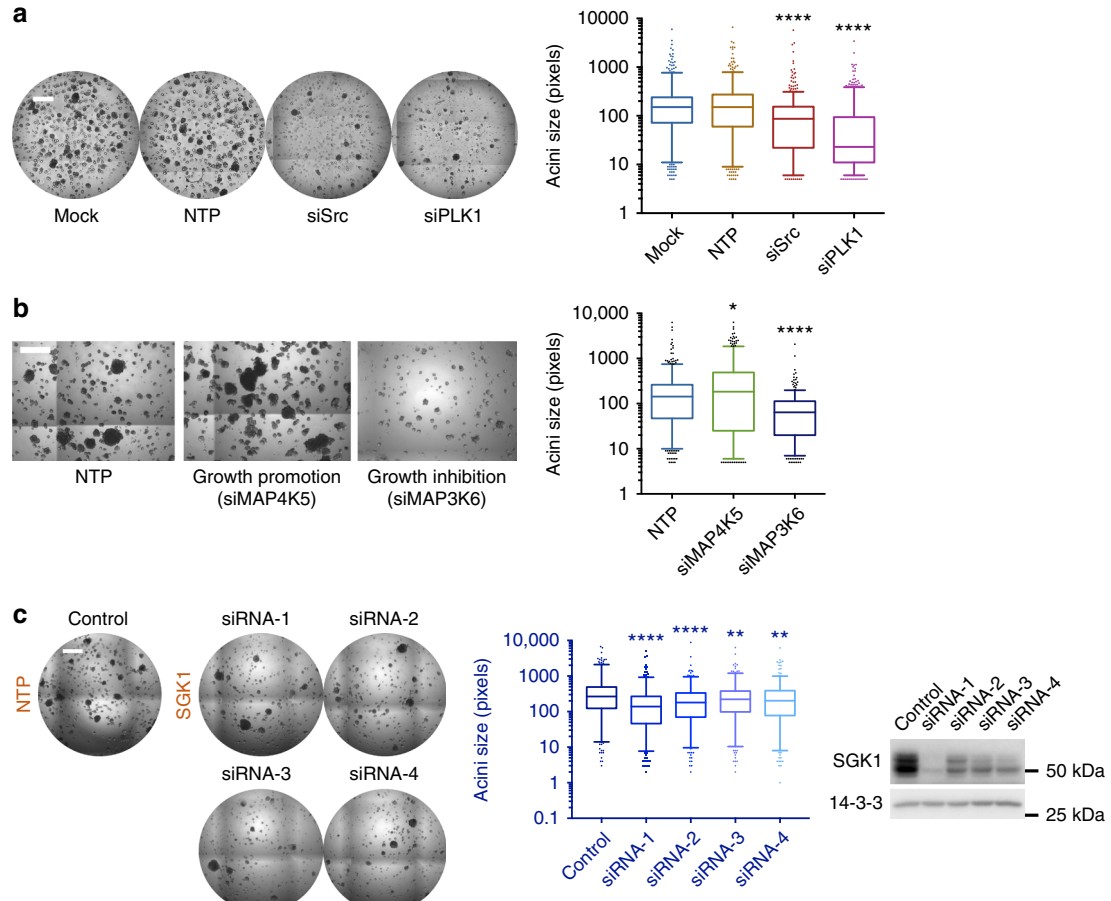

**Fig. 4** Overview of 3D acini formation screen. **a** Optimization of functional screen conditions. 3D growth of MCF-10A_Src cells transfected with siPLK1 (positive control for screen), siSrc (against the expressed Src Y527F) and controls (mock and NTP). Scale bar represents 600 μm. The size of >100 acini were measured and plotted on a box and whisker graph, with center line indicating the median, the upper and lower bounds of box indicating the 25th and 75th percentiles of the data, and the upper and lower whiskers indicating the 5th and 95th percentiles of the data. **b** Primary screen results for MAP4K5 (growth repressor) and MAP3K6 (required for acinar growth). 3D growth of MCF-10A_Src cells transfected with siSMARTpools as indicated. Scale bar represents 400 μm. The quantification method and representation was as for (**a**). **c** Validation of the primary hit SGK1 by SMARTpool deconvolution. 3D growth of MCF-10A_Src cells transfected with individual siRNAs from the SMARTpool (siRNA1-4) (left). Scale bar represents 600 μm. The quantification method and representation (middle) was as for (**a**). Lysates from cells in 2D culture transfected with the same siRNAs were Western blotted as indicated (right). **a–c** For box and whisker graphs *$p < 0.05$, **$p < 0.01$, ****$p < 0.0001$ by Mann–Whitney test

system that would provide temporal information regarding kinome perturbations, and help distinguish direct effects of an oncogene from secondary ones that reflect more general aspects of the transformed phenotype. In addition, while we have validated the Src-mediated regulation and role of SGK1 in additional cell lines, the global impact of active Src on kinase signaling networks and the dependency of Src-transformed cells on specific kinases may vary according to the genetic background of the cells utilized. We also note that the screen was undertaken with a mutant version of Src not characteristic of human cancers. Consequently, it will be interesting to extend our integrated screen beyond the MCF-10A_Src system to cell lines derived from particular cancer types where Src is known to play an important role in disease progression, and interrogate the role of endogenous, wildtype Src in these contexts.

Previous work characterizing the kinomic impact of Src has largely focused on tyrosine phosphorylation events and/or regulation of canonical downstream signaling pathways. The former approach has defined, for example, tyrosine phosphorylation sites on FAK (Y407, Y576 Y577, Y861, Y925) that are regulated by Src and other tyrosine kinases[42,43]. While we detected these modifications in Src-transformed cells, the power of our kinomic

analyses is highlighted by the identification of additional phosphorylation events that likely function in concert with these alterations. For example, we detected decreased phosphorylation on S843 and S722, and each of these sites is known to negatively regulate phosphorylation of the Src recruitment site Y397, and FAK activation[44–46]. Furthermore, while we identified enhanced tyrosine phosphorylation of the RTK EPHA2 in Src-transformed cells, we also detected increased phosphorylation of multiple threonines within the activation loop. Consequently, our approach allows identification of multiple phosphorylation events on a given kinase in response to a specific oncogenic stimulus, and hence important insights into phosphorylation-based signal integration. An additional advantage to expanding the kinomic analysis beyond the tyrosine phosphoproteome is the potential to identify serine/threonine kinases with novel roles in Src transformation. For example, our analyses revealed that MCF-10A_Src cells exhibit decreased phosphorylation of CDK18, a poorly characterized member of the cyclin-dependent kinase family that controls the actin cytoskeleton by negatively regulating FAK activity[31]. The approach also demonstrated complex changes in multisite phosphorylation of MARK1-3. These findings are potentially significant given the known roles of the MARK/Par-1

### Table 1 Validated targets from the functional screen

| Kinase | Kinase group | Proteomics | Fold change | 2D (Reduction) | 3D Validation |
|---|---|---|---|---|---|
| SGK1 | AGC | Expression | **4.02** | < 5% | 4/4 |
| DAPK1 | CAMK | Expression | **2.29** | < 5% | 4/4 |
| PTK2B/FAK2 | TK | Phosphorylation | **Y579(6.17)** | < 5% | 3/4 |
| | | | **Y819(4.76)** | | |
| | | | **Y580(4.85)** | | |
| | | | **Y849(3.48)** | | |
| | | | **S839(1.94)** | | |
| | | | **T842(1.80)** | | |
| SMG1 | Atypical | Phosphorylation | **S3566(1.81)** | 19.6% | 4/4 |
| | | | **T3569(1.54)** | | |
| NEK1 | Other | Phosphorylation | *S418(0.59)* | < 5% | 4/4 |
| SIK3 | CAMK | Phosphorylation | **T859(1.30)** | 23.3% | 3/4 |
| | | | **T922(1.27)** | | |
| MAP3K6 | STE | N/A | N/A | 25.3% | 2/4 |
| MAP4K5 | STE | Phosphorylation | **S335(1.20)** | < 5% | 4/4 |
| | | | **S399(1.44)** | | |
| NEK7 | Other | Phosphorylation | *S188(0.79)* | < 5% | 4/4 |
| LIMK2 | TKL | Phosphorylation | **S291(1.24)** | 6.4% | 3/4 |
| | | | *S297(0.78)* | | |
| | | | *S298(0.76)* | | |

The table lists kinases that markedly perturbed the Src-induced transformed phenotype in 3D culture and were subsequently validated by deconvolution of the corresponding siRNA SMARTpools. The "Proteomics" and "Fold change" columns summarize data from the kinomic screen, indicating whether expression or phosphorylation changes were observed, the corresponding magnitude (in brackets), and the direction, with bold font denoting increased and italic font denoting decreased expression/phosphorylation in MCF-10A_Src cells relative to MCF-10A_Ctrl cells (MS results from Fig. 1a, Supplementary Data 1 and Supplementary Table 1). The "2D (Reduction)" column indicates the reduction in viability upon target knockdown in MCF-10A_Src cells cultured in 2D under −EGF conditions. The "3D Validation" column indicates the number of individual siRNAs from the SMARTpool that perturbed the 3D growth of MCF-10A_Src cells under −EGF conditions. MAP3K6 was included in the functional screen but did not exhibit phosphorylation or expression changes in Src-transformed cells

kinase family in regulating biological endpoints relevant to the transformed phenotype, including cell cycle control, cell polarity and migration[30].

To our knowledge, our functional screen represents the first unbiased attempt to identify the kinase dependencies of Src-induced transformation. Interestingly, this identified a suite of protein kinases with a diverse array of known functions. Aside from SGK1, where we decided to focus our efforts, the identified kinases play roles in processes varying from microtubule organization (NEK1, NEK4[35]), to JNK/p38 MAPK regulation (MAP3K6[47]) and metabolic signaling (SIK3[48]). As described earlier, SGK1 expression is positively correlated with the TNBC subtype. However, the validated functional hits MAP3K6, LIMK2, DAPK1, and PTK2B also exhibit this association (Supplementary Figure 7), further supporting their role as modulators or effectors within the TNBC-associated SFK signaling network. In addition, our data identifying MAP4K5 as a suppressor of Src-induced transformation support an emerging role for this kinase as a negative regulator of tumor progression. Thus, expression of MAP4K5 is decreased or lost in the majority of pancreatic cancers, and low MAP4K5 levels in this malignancy are associated with epithelial-mesenchyme transition and poor prognosis[49]. Consistent with these data, interrogation of publicly-available gene expression data reveals that low MAP4K5 expression is also associated with a worse prognosis in TNBC (Supplementary Figure 12).

Integrated data from the two screens highlighted SGK1 as a strong candidate for further characterization: specifically, expression of this kinase was enhanced in MCF-10A_Src cells, and required for Src-induced transformation in this system. The relevance of SGK1 to the SFK signaling network in TNBC was confirmed by its increased expression in this disease subtype together with its positive regulation by Src in TNBC cells and contribution to colony formation in vitro and tumor growth in vivo. SGK1 regulation by Src, and dependency, was also demonstrated in NSCLC cells, indicating that the identified Src/ SGK1 signaling axis is not specific to TNBC. These data lend

further support to a key oncogenic role for SGK1, which was identified as a driver protein kinase in a recent meta-analysis of cancer genome sequencing studies[2]. Importantly, an emerging characteristic of SGK1 is that it may not only represent a therapeutic target in itself, but also one that can be exploited to improve the efficacy of existing therapies. For example, SGK1 promotes resistance of breast cancer cells to Akt and PI3Kα-directed agents[41,50]. These findings led us to test the combined effect of targeting Src and SGK1 for two reasons. First, in clinical trials, the response of solid malignancies to small molecule Src inhibitors has in general been disappointing, emphasizing the need for identification of predictive biomarkers and/or improved combination therapies. Second, while the expression of SGK1 was positively regulated by Src in TNBC cells, Src inhibition did not totally ablate expression of this kinase, with continuing signal output from the residual SGK1 likely. Consistent with this hypothesis, dual targeting of the two kinases led to enhanced efficacy both in vitro and in vivo, identifying a strategy for improved therapeutic targeting of the SFK signaling network in TNBC and NSCLC. These data highlight how our integrated research strategy not only provides important mechanistic insights into cellular transformation by a specific oncogene, but can also identify biomarkers that predict cellular responsiveness to inhibition of that oncogene, and combination therapies to overcome therapeutic resistance.

## Methods

**Generation of stable cell lines.** MCF-10A cells stably expressing the murine ecotropic receptor (MCF-10A EcoR, a kind gift from Drs. Danielle Lynch and Joan Brugge, Harvard Medical School) were used to generate stable pools of MCF-10A cells expressing Src Y527F (referred as to MCF-10A_Src) or both Src Y527F and SGK1 WT by retroviral-mediated transduction. A pool of pBabe vector expressing MCF-10As (MCF-10A_Ctrl) was used as a control for Src Y527F, and pMIG vector was used as a control for the SGK1 retroviral infection. For the production of stable cell lines, PlatE cells were transfected with constructs using Lipofectamine 3000 (Invitrogen) according to the manufacturer's instructions. Viral supernatants were collected at 48 h and 72 h after transfection and filtered through 0.45-μm pore-size filter caps (Millipore). MCF10-A EcoR cells were infected with viral supernatants for 24 h in presence of 8 μg/ml polybrene (Millipore). Successfully transduced cells

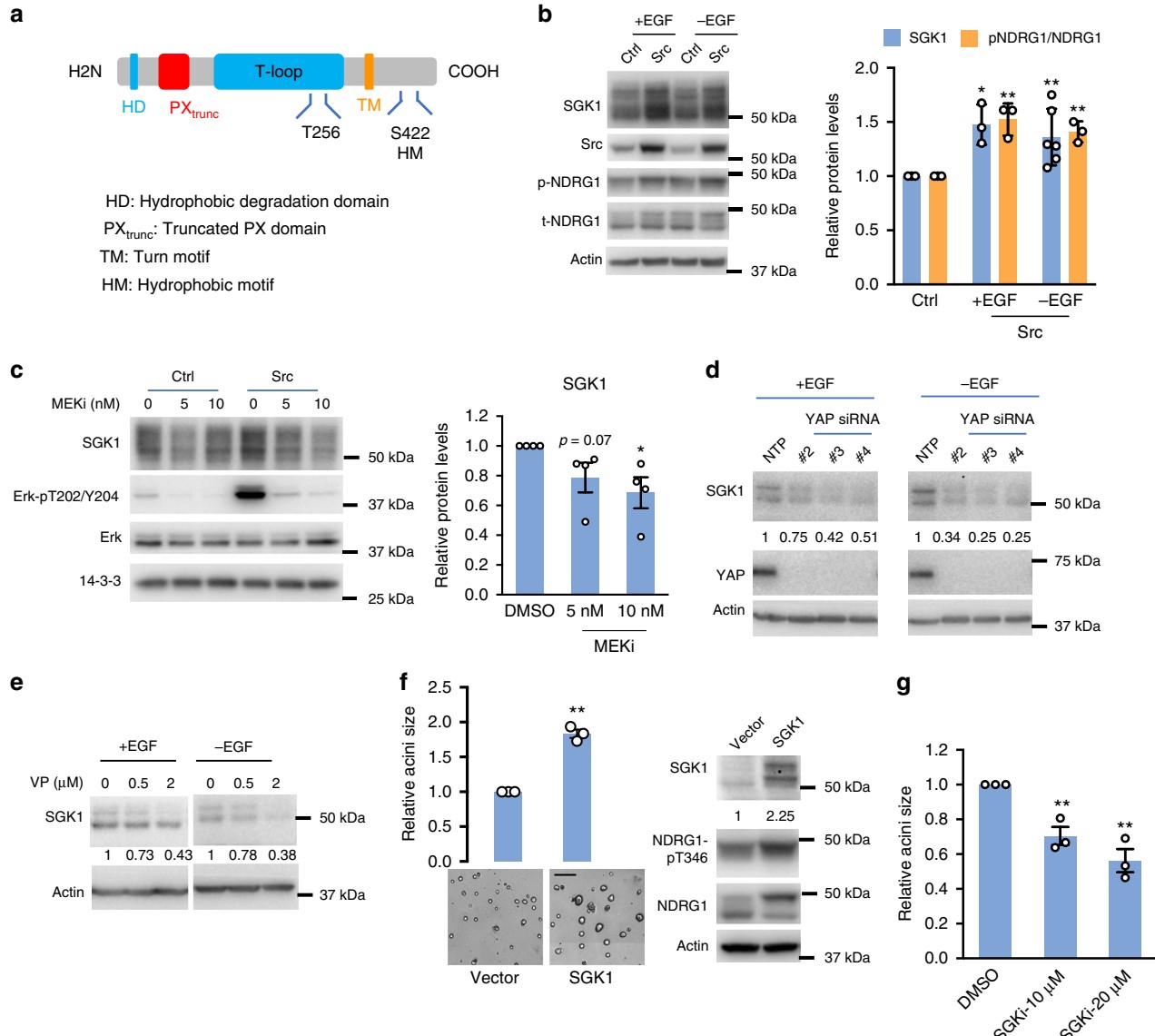

**Fig. 5** SGK1 is required for Src-mediated transformation of MCF-10A cells. **a** Schematic representation of SGK1 protein structure. **b** Src Y527F expression increases SGK1 expression and activity. Lysates from MCF-10A_Ctrl (Ctrl) and MCF-10A_Src (Src) cells cultured in 2D culture −/+EGF were Western blotted as indicated. Representative blots are shown from between $n = 3$ and $n = 6$ biological replicates. **c** SGK1 upregulation by active Src requires MEK activity. Lysates from MCF-10A_Ctrl and MCF-10A_Src cells maintained in −EGF medium in 2D culture and treated with the MEK inhibitor, trametinib for 24 h were western blotted as indicated. DMSO was the vehicle control. Representative blots are shown from four biological replicates. **d** SGK1 upregulation is also YAP-dependent. Lysates from MCF-10A_Src cells cultured in 2D culture −/+EGF and transfected with three individual siRNAs for YAP or non-targeting control (NTP) (25 nM) were western blotted as indicated. **e** YAP inhibition reduces SGK1 expression levels. Lysates from MCF-10A_Src cells maintained as in (**d**) and treated for 16 h with the YAP inhibitor verteporfin (VP) (0.5 and 2 μM) were western blotted as indicated. DMSO was the vehicle control. Representative blots are shown from $n = 2$ biological replicates. **f** SGK1 overexpression increases the size of MCF-10A_Src acini. Flag-SGK1 was overexpressed in MCF-10A_Src cells by retroviral infection and empty vector was used as control. Acini were grown under −EGF conditions and imaged and quantified from $n = 3$ biological replicates. Scale bar represents 200 μm. The right-hand panel shows western blot analysis of cell lysates derived from the 3D cultures. **g** SGK1 inhibition reduces the size of MCF-10A_Src acini. MCF-10A_Src cells were treated with SGK1 inhibitor (SGKi) during acini growth under −EGF conditions. DMSO was the vehicle control. Acini were imaged and quantified from $n = 3$ biological replicates. **b–g** The column graph and numbers below the panels show the quantification of blots (normalized to actin or 14-3-3 which were loading controls) or acini with data expressed relative to the corresponding control which was arbitrarily set at 1. Error bars represent s.e.m., *$p < 0.05$, **$p < 0.01$ by Student's $t$-test

were selected with puromycin (2 μg/mL) for pBabe constructs or sorted by flow cytometry based on GFP expression levels for pMIG constructs.

For inducible knockdown of SGK1, selective shRNAs were cloned into the pLKO-Tet-On backbone. To produce lentiviral supernatants, HEK-293 cells were co-transfected with control or shRNA-containing pLKO-Tet-On vectors, VSVG and psPAX2. Successfully transduced MDA-MB-231 and A549 cells were selected with puromycin (1 μg/mL). Both shRNA sequences are shown in Supplementary Table 4.

**Cell lines and tissue culture**. MCF-10A cells and their derivatives were maintained in Dulbecco's modified Eagle's medium/nutrient mixture F-12 (Invitrogen) supplemented with 5% (v/v) horse serum (Invitrogen), 20 ng/ml human recombinant EGF (R&D Systems), 0.5 μg/ml hydrocortisone (Sigma), 100 ng/ml cholera toxin (Sigma), and 10 μg/ml bovine insulin (Sigma). MDA-MB-468 (catalog no. HTB-132), Hs578T (catalog no. HTB-126), BT549 (catalog no. HTB-122), HCC70 (catalog no. CRL-2315), HCC1954 (catalog no. CRL-2338), BT20 (catalog no. HTB-19), MDA-MB-436 (catalog no. HTB-130), MDA-MB-453 (catalog no. HTB-

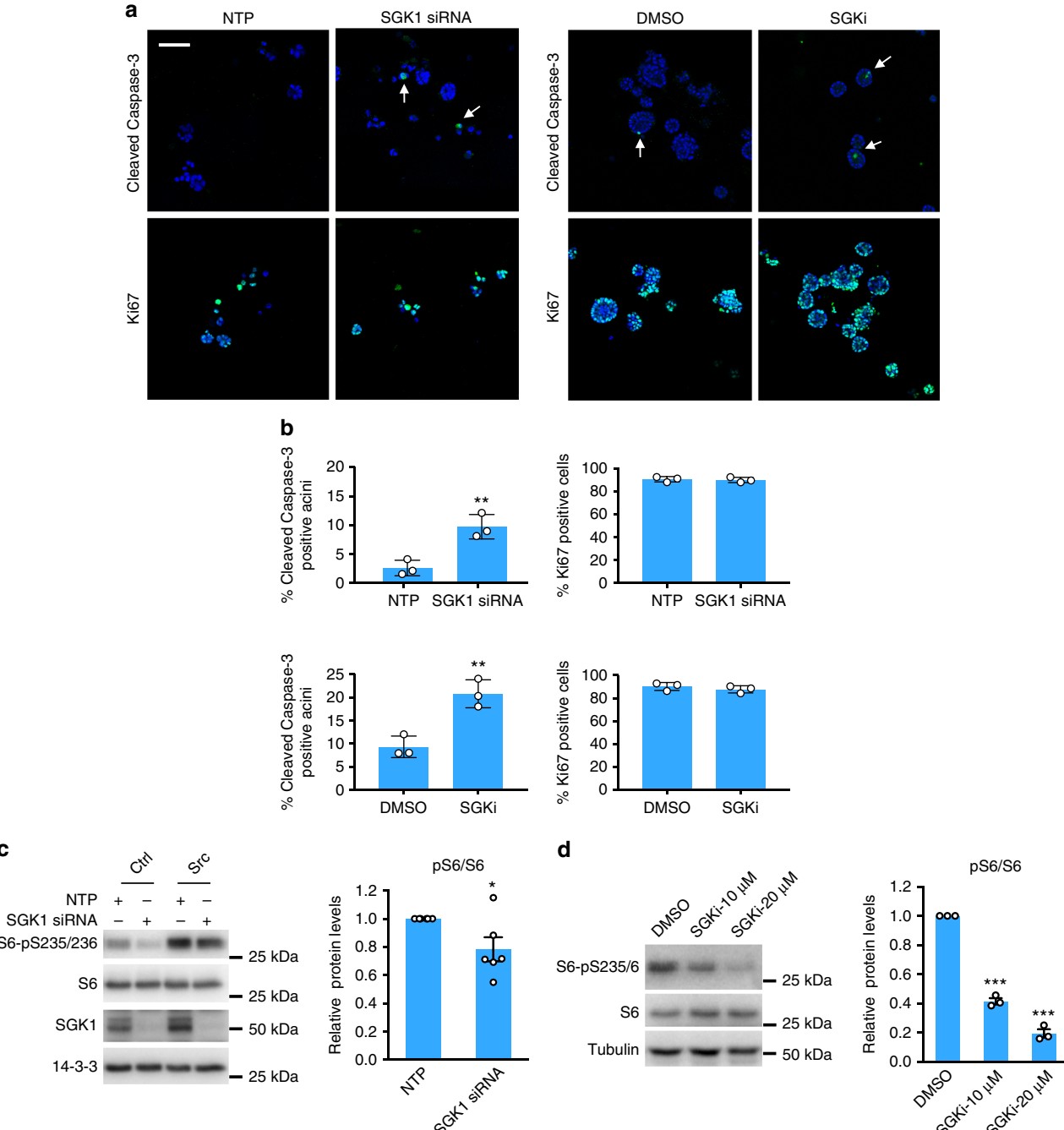

**Fig. 6** Functional characterization of SGK1 in MCF-10A_Src cells. **a** SGK1 knockdown or inhibition increases apoptosis in MCF-10A_Src acini. MCF-10A_Src cells were transfected with a SGK1 siRNA SMARTpool or non-targeting control (NTP) (40 nM) (left) or treated with a selective SGK1 inhibitor (SGKi) (10 μM) (right), and grown in 3D culture for 3 days or 5 days, respectively. Immunofluorescent imaging was performed using Ki67 and cleaved Caspase-3 antibodies (green) and DAPI counterstain (blue). Representative images are shown. Arrows indicate cleaved Caspase-3 positive acini. Scale bar represents 100 μm. **b** Percentage of cleaved Caspase-3 or Ki67 positive acini from $n = 3$ biological replicates. At least 50 acini were analyzed in each replicate. $**p < 0.01$ by Student's $t$-test. **c** Knockdown of SGK1 decreases mTOR activity in 2D culture. MCF-10A_Ctrl and MCF-10A_Src cells were transfected with a SGK1 siRNA SMARTpool or non-targeting control (NTP) (40 nM) and cultured in –EGF medium for 24 h before harvesting. Cell lysates were western blotted as indicated. Representative blots are shown from $n = 6$ biological replicates. The column graph shows quantification of p-S6 normalized to S6, error bars represent s.e.m., $*p < 0.05$ by Student's $t$-test. **d** SGK1 inhibition decreases mTOR activity in 3D culture. MCF-10A_Src cells in 3D culture were treated with SGKi for 24 h and harvested for western blot analysis. Representative blots are shown from $n = 3$ biological replicates. The column graph shows quantification of p-S6 normalized to S6, error bars represent s.e.m., $***p < 0.001$ by Student's $t$-test

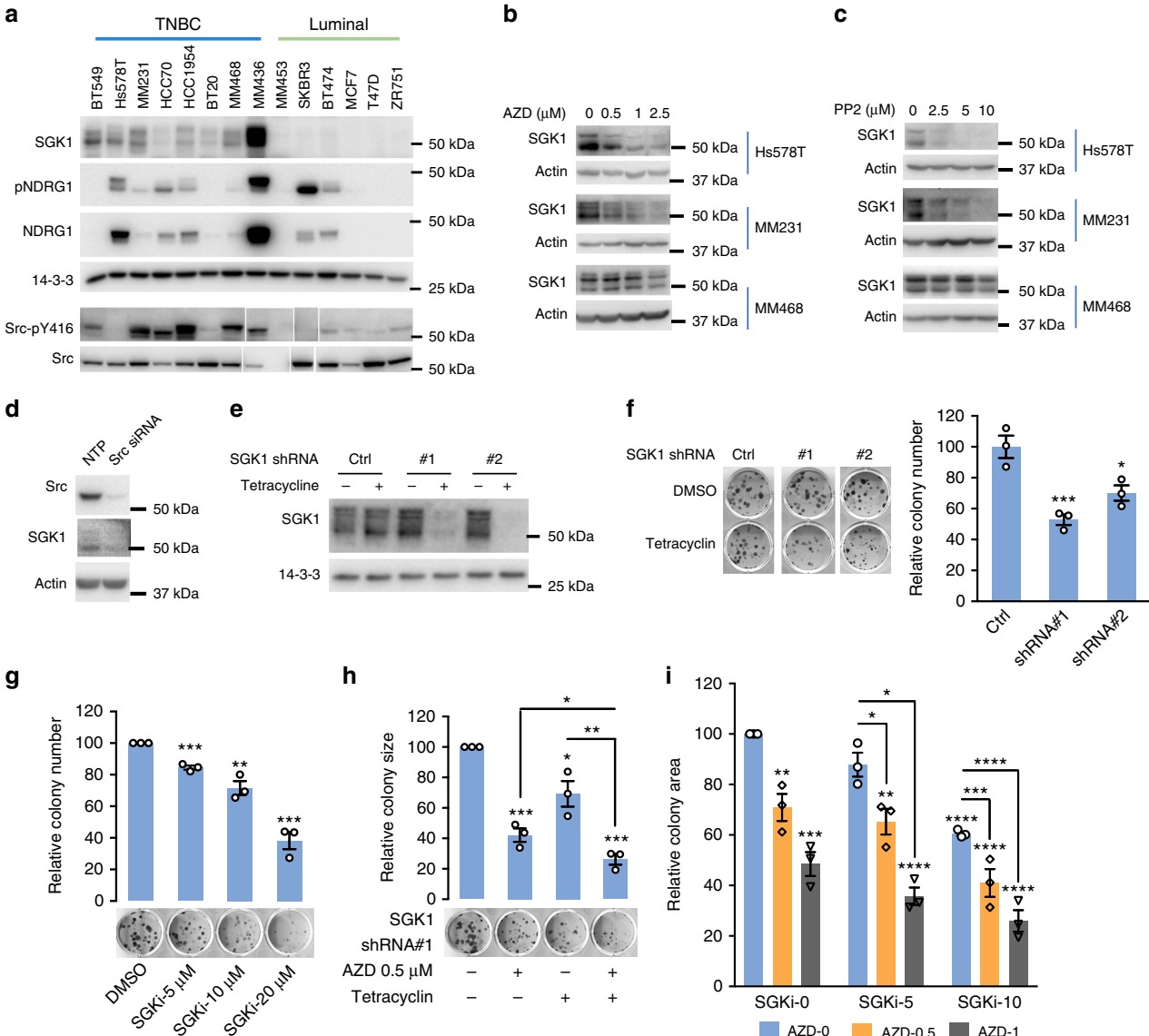

**Fig. 7** SGK1 signals downstream of active Src in TNBC. **a** SGK1 expression and activity in breast cancer cell lines. Cell lysates were western blotted as indicated. The blots for Src and Src-pY416 were run in a different order but have been re-assembled to match the others, for ease of comparison. 'MM' indicates MDA-MB-. **b**, **c** Src inhibition decreases SGK1 expression levels in TNBC cell lines. Lysates were prepared from cells treated overnight with AZD0530 (AZD) (**b**) or PP2 (**c**). DMSO was the vehicle control. Representative blots are shown from $n = 2$ biological replicates. **d** Src knockdown reduces SGK1 expression levels in TNBC. Cell lysates from Hs578T cells transfected with a Src siRNA SMARTpool or non-targeting control (NTP) (20 nM) for 72 h were western blotted as indicated. Representative blots are shown from $n = 2$ biological replicates. **e** SGK1 knockdown by tetracycline-inducible shRNAs. MDA-MB-231 cells were programmed to express tetracycline-inducible SGK1 shRNAs or empty vector (Ctrl) and treated with tetracyclin (1 µg/ml) for 48 h. Cell lysates were prepared and Western blotted as indicated. **f** SGK1 knockdown reduces colony formation by MDA-MB-231 TNBC cells. Cells from (**e**) were used in a colony formation assay. **g** SGK1 inhibition reduces colony formation. MDA-MB-231 cells treated with SGK1 inhibitor (SGKi) or vehicle control (DMSO) were assessed in a colony formation assay. **h** Effect on colony formation of combining SGK1 knockdown with Src inhibitor (AZD) treatment. MDA-MB-231 cells stably transfected with tetracyclin-inducible SGK1 shRNA#1 were treated +/−tetracyclin and +/−AZD. **i** Effect on colony formation of combining the SGK1 inhibitor (SGKi) with Src inhibitor (AZD) treatment. MDA-MB-231 cells were treated +/−SGKi and +/−AZD. Combination index (CI) was calculated by the Chou–Talalay method using the CompuSyn program. The CI value for 10 µM SGKi with 1 µM AZD is 0.8, indicating weak synergy. **f–i** In the column graph, data are expressed relative to the empty vector or DMSO control which was arbitrarily set at 100. Results were quantified from $n = 3$ biological replicates. Error bars represent s.e.m., *$p < 0.05$, **$p < 0.01$, ***$p < 0.001$, ****$p < 0.0001$ by Student's $t$-test

131), SKBR3 (catalog no. HTB-30), BT474 (catalog no. HTB-20), T-47D (catalog no. HTB-133), and ZR-75-1 (catalog no. CRL-1500) cell lines were obtained from the American Type Culture Collection. MDA-MB-231 and MCF7 cell lines were obtained from EG&G Mason Research Institute, Worcester, MA. Breast cancer cell lines were maintained in RPMI-1640 medium (Life Technologies, catalog no. 11875119) supplemented with 10% FBS (Fisher Biotec, catalog no. S-FBS-US-015,

lot no. ASM1-137A11), 10 µg/mL Actrapid penfill insulin (Clifford Hallam Healthcare), and 20 mM HEPES (Life Technologies, catalog no. 15630080). MCF-10A_Ctrl and MCF-10A_Src cells used for the SILAC experiments were cultured for at least six passages in media containing either L-arginine and L-lysine (light) or arginine-13C6-15N4 and lysine-13C6-15N2 (heavy) at final concentrations of 18.43 mg/L (Arg) and 91.25 mg/L (Lys). A549 cells were

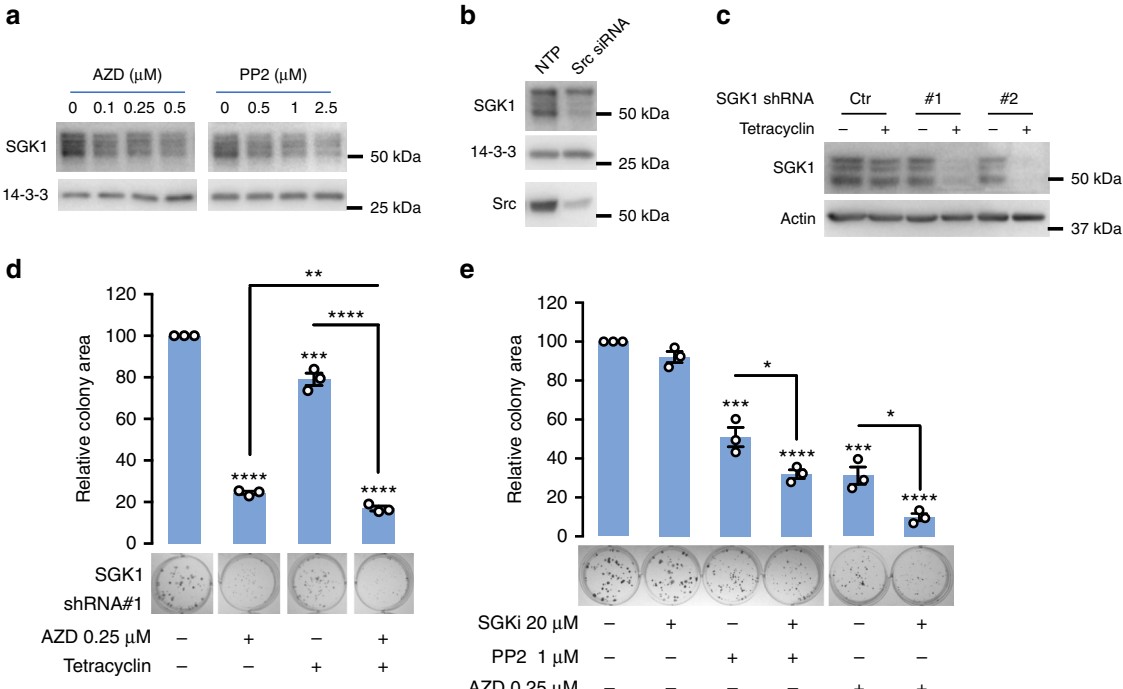

**Fig. 8** SGK1 signals downstream of active Src in non-small cell lung cancer. **a** Src inhibition decreases SGK1 expression levels. Lysates from A549 cells treated for 16 h with AZD0530 (AZD) or PP2 were western blotted as indicated. Representative blots are shown from $n = 2$ biological replicates. **b** Src knockdown decreases SGK1 expression levels. A549 cells were transfected with a Src siRNA SMARTpool or non-targeting control (NTP) (20 nM) and cell lysates harvested at 72 h for western blot analysis. **c** SGK1 knockdown by tetracycline-inducible shRNAs. A549 cells stably programmed to express tetracycline-inducible SGK1 shRNAs or empty vector (Ctrl) were treated with tetracyclin (1 μg/ml) for 48 h and lysates were collected for western blot. **d** Effect on A549 colony formation of combining SGK1 knockdown with Src inhibitor (AZD) treatment. A549 cells stably transfected with tetracyclin-inducible SGK1 shRNA#1 were treated +/−tetracyclin and +/−AZD. Data are expressed relative to the DMSO control which was arbitrarily set at 100. **e** Effect on colony formation of combining SGK1 inhibitor (SGKi) with Src inhibitor (AZD) treatment. A549 cells were treated +/−SGKi and +/−AZD. Data are expressed relative to the DMSO control which was arbitrarily set at 100. Coefficient of drug interaction (CDI) is 0.68 (SGKi and PP2) and 0.34 (SGKi and AZD), respectively, indicating synergy. **d, e** Results were quantified from $n = 3$ biological replicates. Error bars represent s.e.m., $^*p < 0.05$, $^{**}p < 0.01$, $^{***}p < 0.001$, $^{****}p < 0.0001$ by Student's $t$-test

obtained from Professor Jian Li (Monash Biomedicine Discovery Institute) and were cultured in DMEM supplemented with 10% (v/v) FBS (Fisher Biotech, catalog no. S-FBS-US-015, lot no. ASM1-137A11). All cell lines were authenticated by short tandem repeat polymorphism, single-nucleotide polymorphism, fingerprint analyses and underwent routine mycoplasma testing by PCR.

**MS-based kinomic profiling.** For large-scale kinome enrichment and phospho-peptide enrichment from MCF-10A_Src cells and controls, kinases were enriched from 100 mg of light and heavy-labeled protein mixture using 2 ml of kinobead cocktail containing immobilized Purvalanol B (Tocris, UK), CTx-0294885 (also known as KiNet-1, SYNkinase, Australia), SU6668 (Biochempartner Chemical, China) and VI16832 (Evotec, Germany). The resin-bound proteins were eluted and separated into two fractions, for in-gel and in-solution tryptic digestion, respectively. Ten percentage of the in-gel extracts were subsequently used for total protein identification. The remaining 90% and all peptides yielded from the in-solution digestion were subjected to phosphopeptide enrichment using TiO$_2$ (GL Sciences, Japan). The peptide mixture was desalted and concentrated on homemade C$_{18}$ Stage Tips.

Peptides were separated by nano-LC using an Ultimate 3000 HPLC and autosampler system (Dionex) and mass spectra were acquired on an Orbitrap Velos (Thermo Electron) for the MCF-10A_Src profiling and on a Q-Exactive Plus-Orbitrap (Thermo Scientific) for the MDA-MB-231 profiling. Samples were concentrated and desalted onto a micro C18 pre-column (500 μm × 2 mm, Michrom Bioresources) with 2% ACN in 0.05% TFA at 15 μl/min for 4 min. The pre-column was then switched on-line with a nano-C18 column (75 μm i.d. × ~10 cm, 5 μm bead size, 200 Å Magic, Michrom) and the reverse phase nano-eluent was subject to nano-flow electrospray analysis at 250 nl/min for 60 min in a data dependent acquisition mode.

For the Velos, the survey scan (MS1) was acquired at 60,000 resolution from 300 to 1750 $m/z$ with lock-mass enabled. Up to 15 most abundant ions (minimum signal threshold of 2000, charge state 1+ excluded) were sequentially isolated and further subjected to MS/MS (MS2) fragmentation within the linear ion trap using normalized collision-induced dissociation and dynamically excluded for 15 s.

Data acquired on the Q-Exactive consisted of a survey scan (MS1) at 70,000 resolution (automatic gain control target 1e6 and maximum injection time of 50 ms) from 375 to 1800 $m/z$. Up to 10 most abundant ions (charge state 1+, unassigned, >+6) were sequentially isolated and fragmented in a higher energy collisional dissociation (HCD) cell at normalized collision energy (NCE) 27 followed by tandem MS2 scans at 17,500 resolution (automatic gain control target 1e5 and maximum injection time of 120 ms) and dynamically excluded for 15 s.

Raw files were processed with MaxQuant software (v1.2.2.5, 1.3.0.5, and 1.5.2.8) for feature detection, protein identification and quantification and searched against UniProtKB/Swiss-Prot Homo sapiens (v2010_10, 35,052 entries, v2015_08, 20,210 entries) customized chicken Src sequence, and reverse decoy database) using the Andromeda search engine. Searches were performed with a precursor mass tolerance set to 20 ppm, fragment mass tolerance set to 0.5 Da, minimum peptide length set to six amino acids, enzyme specificity set to Trypsin/P, and a maximum number of missed cleavages set to 2. Static modifications were limited to cysteine carbamidomethylation, as well as Arg 0/Lys 0 and Arg 10/Lys 8 for SILAC-labeled samples, and variable modifications used were methionine oxidation, N-acetylation, and phosphorylation of serine, threonine and tyrosine. For data acquired on the Q-Exactive, fast label-free quantification (LFQ) with a minimum ratio count of 2 was performed. The 'match between runs' option in MaxQuant was used to transfer identifications between runs based on matching of precursors with high mass accuracy[51]. Peptides were further filtered using FDR < 0.01, posterior error probability <0.10, and a minimum of one unique peptide. For phosphopeptides, those exhibiting a phosphosite localization probability (LP) >0.75 and score difference ≥5 were included in further analyses. The fold change values were obtained from triplicate independent kinomic profiling experiments, each including a label swap where either the control or MCF-10A_Src cells were heavy-labeled, providing six biological replicates. An appropriate cut-off was determined using the same approach as described previously[52], but more stringent criteria. The frequency distribution of the phosphosite fold changes was plotted and the data divided into 10 terciles with different percentage cut-offs. A fold change of 1.5-fold was selected as the 135 phosphosites defined by this are within the top and bottom 10% of all phosphosites (10th and 90th percentile), respectively. In addition, one sample $t$-test statistical analysis of replicates indicated that only 17 of these sites

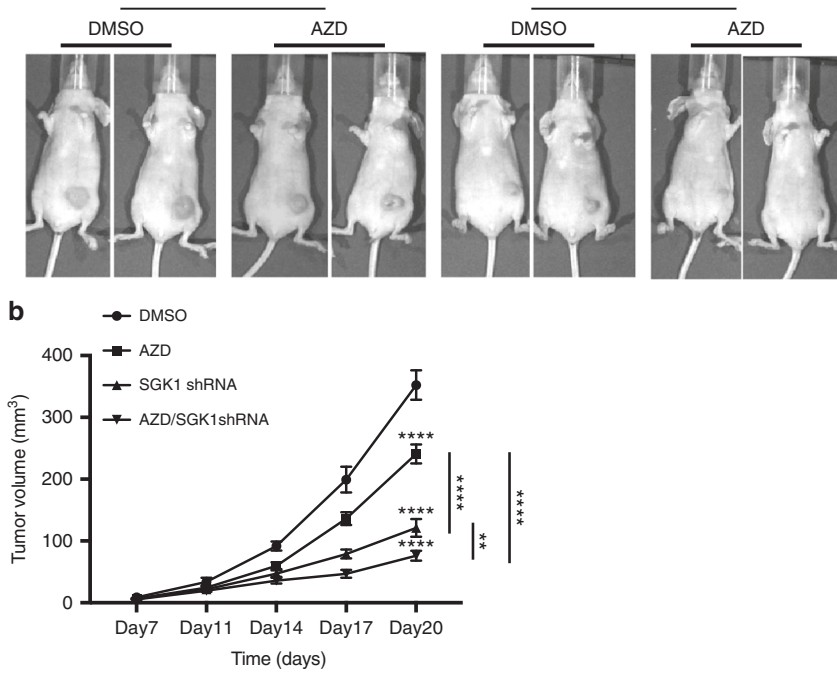

**Fig. 9** Effect of combining SGK1 knockdown and Src inhibitor treatment on tumor growth. **a** Representative images of tumors. $2 \times 10^5$ MDA-MB-231 cells stably transfected with tetracyclin-inducible SGK1 shRNA#2 or empty vector control were injected orthotopically into the fourth mammary fat pad of BALB/c nude mice. Doxycyclin and AZD0530 (AZD) treatment was started 1 day later. Doxycycline was administered in the food and AZD was administered by oral gavage daily at 50 mg/kg. Tumors were allowed to grow for 20 days. Representative images of two tumors for each group are shown. **b** Mean tumor volume from (**a**). Tumor volumes were generated from seven (Ctr) or eight (other three groups) mice. Error bars represent s.e.m., **$p <$ 0.01, ****$p <$ 0.0001 by two way ANOVA

exhibited a *p* value >0.05, and for these, the direction of change for all replicates was consistent (i.e. an increase for upregulated sites, and a decrease for downregulated sites). In order to avoid false negatives, we have retained these 17 sites within the dataset but highlighted them (Supplementary Data 1).

For kinomic profiling of MDA-MB-231 cells treated with AZD0530, cells were treated with the drug (1 μM) or vehicle control for 16 h, and then subjected to the above MS workflow.

**Generation of the kinome tree**. The Kinome tree illustration was reproduced courtesy of Cell Signalling Technology Inc. (www.cellsignal.com) and annotated using Kinome Render software (available at http://bcb.med.usherbrooke.ca/kinomerender)[53].

**Construction of protein–protein interactome networks**. The annotations of protein–protein interactions for all kinases in this study were extracted from the STRING database[54] by using the database search function of Cytoscape software[55]. For a few protein interaction partners whose annotations could not be extracted, we manually added them to the protein–protein interaction networks. Then, significantly upregulated and downregulated kinases within the context of STRING interactome networks were visualized and highlighted using Cytoscape at both the protein and phosphosite levels.

**High-throughput screen for monolayer (2D) proliferation**. Primary screening experiments were conducted independently in triplicate for both control and Src-transformed MCF-10A cells. Cells were distributed into white walled clear bottom 96-well plates (catalog no. 3610, Corning) at $4 \times 10^3$ cells/well for the +EGF condition (full medium, Dulbecco's modified Eagle's medium/nutrient mixture F-12 supplemented with 5% (v/v) horse serum, 100 ng/ml cholera toxin, 0.5 μg/ml hydrocortisone, 10 μg/ml bovine insulin, 20 ng/ml EGF), and $6.5 \times 10^3$ cells/well for the −EGF condition (Dulbecco's modified Eagle's medium/nutrient mixture F-12 supplemented with 0.4% (v/v) horse serum, 100 ng/ml cholera toxin, 0.5 μg/ml hydrocortisone, 10 μg/ml bovine insulin) using liquid handling automation (Bio-Tek 406), and reverse transfected using robotic automation (Caliper Sci Clone ALH3000) with 40 nM siGENOME SMARTpool siRNA using DharmaFECT 3 (Dharmaon RNAi Technologies). Media was changed after 24 h for both conditions and at 72 h post-transfection for the −EGF condition. Cell viability was determined using the Cell Titer-Glo luminescence assay (Promega, Australia) following standard procedures but working with a 1:4 dilution. All luminescent

measurements were taken using the Synergy H4 high-throughput multimode microplate reader (BioTek, USA).

**Three-dimensional (3D) cultures of MCF-10As**. To establish 3D cultures, cells were trypsinized and resuspended in 3D –EGF medium (Dulbecco's modified Eagle's medium/nutrient mixture F-12 supplemented with 2% (v/v) horse serum, 100 ng/ml cholera toxin, 0.5 μg/ml hydrocortisone, 10 μg/ml bovine insulin) or 3D +EGF medium (Dulbecco's modified Eagle's medium/nutrient mixture F-12 supplemented with 2% (v/v) horse serum, 100 ng/ml cholera toxin, 0.5 μg/ml hydrocortisone, 10 μg/ml bovine insulin, 5 ng/ml EGF), and plated into a 96-well plate pre-coated with 40 μl of Matrigel at a density of ~800 cells/well in medium containing 2% Matrigel. Cells were allowed to form acini for up to 11 days and fresh medium was replaced every 3 to 4 days.

**Functional screen for 3D proliferation**. MCF-10A_Src cells were reverse transfected with individual siRNA SMARTpools using DharmaFECT 3 in a 96-well plate. One day post-transfection, cells were trypsinised and seeded onto Matrigel as described in the 3D model above. Cells were allowed to form acini for up to 11 days and fresh medium was replaced every 3 to 4 days. During 3D cell imaging, each well was first imaged at different focal points in a $3 \times 4$ tiled format under a 4× objective using the Leica AF6000LX Live Cell Imaging workstation, followed by tile stitching using Las X software and layers stacking using ZereneStacker. Acinar size was measured using ImageJ (Version 1.51).

**Indirect immunofluorescence of acini**. Acini structures were fixed and assessed by indirect immunofluorescence. Briefly, acini were fixed with 2% paraformaldehyde in PBS for 20 min and permeabilised with 0.05% Triton X-100 in acini-PBS (130 mM NaCl, 7 mM Na$_2$HPO$_4$, 3.5 mM NaH$_2$PO$_4$, pH 8) for 10 min at 4 °C. Acini were washed three times with 100 mM glycine in acini-PBS for 15 min each. Acini were blocked for 1.5 h with primary block solution containing IF buffer (acini-PBS, 7.7 mM NaN$_3$, 0.1% bovine serum albumin, 0.2% Triton X-100, 0.05% Tween-20) with 10% goat serum (Gibco), then blocked for a further 40 min in secondary block solution (primary block solution with 20 μg/mL goat anti-mouse F (ab)'$_2$ fragment (Jackson ImmunoResearch, catalog no. 115-006-006)). Primary antibodies were added in primary block solution overnight at 4 °C, then acini were washed three times with IF buffer for 20 min each with gentle rocking. Alexa Fluor® secondary antibodies and DAPI were added in primary block solution for 45 min at room temperature. Acini were washed three times with IF buffer for 20 min each with rocking. Acini were washed once with acini-PBS, then mounted

onto cover slips with Fluoromount G (Electron Microscopy Services, catalog no. 1798425). Images were obtained using a Leica SP8 invert confocal laser scanning microscope.

**Colony formation assay**. Cells were plated in 12-well plates at 100 cells/well, and treated with inhibitors/drugs or DMSO control 24 h later. Media was replaced every 2 days with fresh inhibitors/drugs until colonies were evident. Cells were then fixed with methanol and colonies stained with 0.4% crystal violet. Colony number and colony size were quantified with ImageJ (Version 1.51).

**Transfections**. SMARTpool siRNA or individual siRNA duplexes were applied to cells using DharmaFECT (Dharmacon, Lafayette, CO) transfection reagent according to the manufacturer instructions. SiRNA sequences and catalog numbers are shown in Supplementary Table 5.

Plasmid transfections were performed using Lipofectamine 3000 (Invitrogen) according to the manufacturer instructions. pBabe, pBabe-Src Y527F and pMIG vectors were as utilized in previous work[20,56]. pMIG/SGK1 WT was purchased from GenScript.

**Immunoblot**. Cell lysates were prepared for immunoblotting using 2% SDS in PBS. Western blot analysis was performed following SDS-polyacrylamide gel electrophoresis using 5% stacking gels and 8% or 10% separating gels. Fully uncropped scans of all blots presented in the main figures are provided in Supplementary Figure 13.

**Antibodies**. Antibodies against beta-actin (1:5000, catalog no. sc-69879), Pan 14-3-3 (1:10,000, catalog no. sc-1657), and MAP4K5 (KHS, 1:500, catalog no. sc-6429) were purchased from Santa Cruz Biotechnology (Dallas, TX). α-tubulin (1:5000, catalog no. T5168) was purchased from Sigma. SGK1 (1:1000, catalog no. 3272 and 12013), pErk-T202/Y204 (1:2000, catalog no. 4370S), Erk (1:2000, catalog no. 4695S), pJNK-T183/Y185 (1:1000, catalog no. 9252), JNK (1:2000, catalog no. 9251), p-p38-T180/Y182 (1:1000, catalog no. 4511), p38 (1:2000, catalog no. 9212), pYAP-S127 (1:2000, catalog no. 4911), YAP (1:2000, catalog no. 14074), pNDRG1-T346 (1:2000, catalog no. 5482), NDRG1 (1:2000, catalog no. 9485), pS6-S235/236 (1:1000, catalog no. 2211), S6 (1:1000, catalog no. 2217), pGSK3α/β-S21/9 (1:1000, catalog no. 9331), pSrc-Y416 (1:1000, catalog no. 2123), and cleaved Caspase-3 (Asp175) (1:100 for IF, catalog no. 9664) were purchased from Cell Signaling Technology (Danvers, MA). Ki67 (1:200 for IF, catalog no. RM-9106) was purchased from Thermo Fisher. Donkey anti-rabbit Alexa Fluor® 488 (1:500, catalog no. A-21206) was purchased from Life Technologies.

**Inhibitors and chemicals**. The Src inhibitor AZD0530 (catalog no. S1006) was purchased from Selleck, USA. SGK1 inhibitor (catalog no. RGNCY-0058) was purchased from Reagency (Vic, Australia). Rapamycin (catalog no. R0395), PP2 (catalog no. P0042), Verteporfin (catalog no. SML0534) and DAPI (1:500, catalog no. D9542) were purchased from Sigma. MEK inhibitor Trametinib (catalog no. HY-10999) was purchased from MedChem Express (Monmouth Junction, NJ).

**Quantitative real-time PCR**. The RNeasy minikit (Qiagen GmbH, Germany) was used to extract total RNA. Reverse transcription was conducted with the High-Capacity cDNA Reverse Transcription Kit (Applied Biosystems, catalog no. 4368814). Quantitative real-time PCR (qPCR) was carried out on an C1000 Touch™ Thermal Cycler qPCR machine (Bio-Rad) using FastStart Essential DNA Green Master (Roche, catalog no. 6402712001). Data were analyzed through the $2^{-\Delta\Delta Ct}$ method and normalized to the GAPDH housekeeping gene. Primers are shown in Supplementary Table 4.

**Xenografts**. All procedures involving mice were conducted in accordance with National Health and Medical Research Council (NHMRC) regulations on the use and care of experimental animals and the study protocol approved by the Monash University Animal Ethics Committee. $2 \times 10^5$ MDA-MB-231 cells stably transfected with tet-on SGK1 shRNA#2 or empty vector were suspended in 20 µl PBS and injected into the fourth mammary fat pad of 6- to 8-week-old female BALB/c athymic mice purchased from Animal Resources Centre (Canning Vale, Australia). One day post-injection, Doxycycline was administered in the food (600 mg doxycycline/kg) (Specialty Feeds, Australia) and AZD0530 was administered by oral gavage daily at 50 mg/kg until the end of the experiment. Tumor width and length were measured using digital calipers every 2 to 3 days. Tumor volume was determined using the formula ½ × length × width$^2$ = volume of an ellipsoid in mm$^3$. Mice were imaged using a Xenogen IVIS200 system. At the end of the experiment, the animals were humanely killed by $CO_2$ asphyxiation and cervical dislocation, and Xenografts were collected. Results are presented as mean +/− SEM of tumor volume.

**Statistics and synergy calculations**. Two-tailed Student's *t*-test, Mann–Whitney test, ANOVA were performed as indicated in figure legends to determine statistical significance. Combination indices (CI) were generated by the Chou–Talalay

method using the CompuSyn program[57]. Coefficients of drug interaction (CDIs) were calculated as follows: $CDI = E_{AB}/(E_A \times E_B)$, where $E_{AB}$ is a normalized biological response at combination treatment of Drug A and Drug B, and $E_A$ and $E_B$ are the response measured at single drug treatment, respectively. CDI < 1, =1 or >1 indicates that the drugs are synergistic, additive or antagonistic, respectively.

**Reporting summary**. Further information on experimental design is available in the Nature Research Reporting Summary linked to this article.

## Data availability

The mass spectrometry proteomics data have been deposited to the ProteomeXchange Consortium via the PRIDE[58] partner repository with the dataset identifier PXD010687. Gene expression data were extracted from Breast Cancer Gene Expression Miner 4.0 (http://bcgenex.centregauducheau.fr). A Reporting Summary for this Article is available as a Supplementary Information file. All other data supporting the findings of this study are available from the corresponding author on reasonable request.

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

## Acknowledgements

R.J.D. is a NHMRC Principal Research Fellow (APP1058540). The Victorian Centre for Functional Genomics (K.J.S.) is funded by the Australian Cancer Research Foundation (ACRF), the Australian Phenomics Network (APN) through funding from the Australian Government's National Collaborative Research Infrastructure Strategy (NCRIS) program and the Peter MacCallum Cancer Centre Foundation. We acknowledge the scientific and technical assistance of Dan Thomas and Jennii Luu from the VCFG. We acknowledge the facilities, scientific and technical assistance of Monash Micro Imaging and Monash Biomedical Proteomics Facility, Monash University, Victoria, Australia.

## Author contributions

Conceptualization: R.D; Methodology: R.D., L.Z., X.M., J.S., J.W., E.N., S.R., R.S., K.B., C.M., and K.S.; Investigation: L.Z., X.M., E.N., R.L., S.R., F.L., C.H., H.C., and C.C.; Writing original to final draft: R.D., X.M., and L.Z. with input from J.S., R.L., and Brock J. Conley; Review and editing: E.N., R.L., K.B., and K.S.; Funding acquisition: R.D.; Project administration and supervision: R.D. and K.S.

## Additional information

**Competing interests:** The authors declare no competing interests.

