## [Peer Review File · Nature Communications]

Reviewers' comments:

Reviewer #1 (Remarks to the Author):

The main part of this study is MS-based chemical proteomics in normal and Src-transformed mammary epithelial cells. Using this unbiased system, the authors characterized the global impact of Src transformation on the expressed kinome, including non-tyrosine phosphorylated kinases, and functionally annotate the regulated kinases. The MS analysis identified protein kinases with altered expression and/or phosphorylation in Src-transformed cells, and siRNA screening identified nine kinases, including SGK1, that are essential for Src transformation. Interestingly, they also found that phosphorylation of MAP4K5 rather suppressed transformation. To validate the role of SGK1 in human cancers, they showed in triple negative breast cancer cells that Src positively regulated SGK1 expression and combined inhibition of Src and SGK1 effectively inhibited cell proliferation. Based on these findings, the authors propose that this approach not only provides major mechanistic insights into oncogenic transformation but also aids the design of improved therapeutic strategies.

Generally, this study technically sounds and contains interesting findings such as the identifications of SGK1 as an important downstream factor of Src and MAP4K5 as a suppressor of Src transformation. However, I have some concerns that need to be addressed by the authors before publication.

Major points:

- 1) For MS-based kinome analysis, the authors used MCF10 A and Src-transformed MCF10 A. However, since Src gene is not necessarily a driver proto-oncogene (there is no significant Y527F mutation in any cancers), this model system may not be suitable for functional annotation of the Src downstream kinases in actual cancer. The authors should use some cancer cells that have substantial upregulation of Src protein, such as A549 lung cancer cells. By comparing these cancer cells with those in which Src function is attenuated by shRNA or some specific inhibitors, the MS-based analysis should be performed. An inducible system may also be useful for shRNA analysis. Furthermore, the authors should add more appropriate explanation for the advantage of MCF10A-Src system. If they need to stick to this system, they should employ inducible systems for WT and Y527F Src to address the role of Src signaling at differential stages of cancer progression.
- 2) The authors identified MAP4K5 and SGK1 as representative downstream effectors of Src transformation. However, the role of MAP4K5 remains unclear. Although overexpression and siRNA knockdown induced tumor suppressive effects in Src-transformed MCF10A cells, the effects of overexpression of phosphomimetic mutant are limited. Is MAP4K5 really phosphorylated at S335 in cells? If so, what kinds of kinases phosphorylate MAP4K5? What is the stoichiometry of the S335 phosphorylation? To demonstrate the important role of MAP4K5, more detailed analysis is required. If it is difficult to do so, the part of MAP4K5 should be deleted to make the story more simple.

- 3) MS data show that phosphorylation of SGK1 at S401 is increased only 1.23-fold in Src transformation. The phosphorylation at this site should be verified by other techniques, such as Western blotting with a specific antibody against phospho-S401. Also, Src-induced upregulation of SGK1 protein is no more than 1.5-fold. To support the potential contribution of SGK1 to Src transformation, effects of upregulation of SGK1 on acini formation in control MCF10A cells need to be examined. If SGK1 were a critical driver, it could transform MCF10A cells in a manner like Src.
- 4) Fig. 8f. To propose the effectiveness of combination therapies using inhibitors of Src and SGK1, the effects of combined treatment with AZD and SGK1 should be examined instead of the combination of SGK1 knockdown and AZD0530.
- 5) Suppl. Fig.2. could be made more easy to read.

Reviewer #2 (Remarks to the Author):

In this work, the authors compare the kinomes of SRC transformed cells vs those that are not in order to identify factors that are involved in the transformation process. Although the study is interesting from the point of view that many aspects of the transformation process are not clear, the work has a number of issues that diminish its value.

the kinome enrichment was performed 3 times but it remains entirely unclear how the authors arrive at their chosen cut-off values above/below which they consider kinase to change.

the chosen cut-offs are tiny, asking the question if such small differences can be robustly detected by SILAC labeling and a triplicate analysis. At the very least, one would expect the cut-offs to be different for different kinase expression/purification levels.

a worrying statement is that the number of p-sites that are regulated up or down are very similar. This may well imply that simple technical variation is in the same order.

in addition, they imply that differential kinome enrichment on the resin that they used implies differential activity of the kinases. This may not at all be justified.

the authors claim that their approach is novel but it is not. They only cite their own work on the CTx cpd but fail to acknowledge several labs that have come up with similar beads before.

page 5: a third of phosphorylation regulation on SP sites can hardly be called SRC-specific

large parts of the manuscript are descriptive and we do not learn how these relate to SRC transformation. This reviewer suggests to shorten the descriptive part and focus on the functional validation

Speaking of which: Fig 4a is not very clear. Has this data been corrected for multiple testing? How do the kinases relate to each other under the two conditions? in this visualisation, it is unclear which kinase really show a clear difference.

siRNA data: which of the hits that scored in the siRNA screen also showed regulation in the proteomic experiment? this would seem an important piece of information in order to assess the quality of the initial experiment

Fig 7f: how do the authors know that SGK1 cpd is selective? it is used at 10 and 20 uM which are very high concentrations for kinase inhibitors and at which concentration almost no KI is selective

same thing for AZD0530

Fig1: the data implies that something is going on on Tyr kinases. The rest is speculation and likely just owing to rather too optimistic cut-offs from the proteomic experiments

Fig 2 does not add anything in particular to the manuscript

Fig 3a: can the authors explain why the SRC transformed cells also stop growing as do the control cells (albeit at a lower cell density)? would one not expect cell growth just to be slower but eventually go higher than 25% of the controls?

Fig 3c is superfluous. How relevant is it to inspect cells after 7 days or 11 day?

Fig 4a: we don't learn anything from this figure. it would be important to connect the individual kinases between control and SRC transformation.

Figure 4b uses arbitrary cuts which leads to overinterpretation of the data (essential for both etc)

Fig 5b: Given the huge variation in the data, it is unclear if the statistical test actually means anything. Same in Fig 5d. Statistical significance may simply be reached because there are so many data points.

Fig. 6f is not very convincing. why should the Ala mutant have no effect but the phospho-mimetic a clear effect? The Ala mutant should show the opposite effect.

Fig 7f/g: how can the authors assume their SGK1 inhibitor is selective at the very high concentrations applied? More controls would be required to establish a role of this kinase in this context.

#

Reviewer #3 (Remarks to the Author):

The authors describe their approach to discovering novel downstream events caused by expression of active Src kinase and associated Src-mediated transformation, by predominantly using MCF-10A cells that express a constitutively activated Src variant. Essentially, they perform kinome profiling – expression and phosphorylation changes - between Src-expressing and wild-type cells. This gives an extensive list of changes that implicates some potential effectors or suppressors of Src transformation, and they use it to focus on an siRNA screen of altered kinases in a 2D viability assay and a 3D transformation assay. They focus in on a potential suppressor kinase, MAP4K5, and a promoting kinase, SGK1 – providing some evidence for their involvement. Overall, I think this work has merit and it does reveal potentially novel effectors of Src-mediated event in cancer. However, enthusiasm for publishing this work as presented is dampened by a number of general weaknesses; 1) the discovery work was done in only one cell line (was it only one clone of Src-transformed MCF-10A cells and one WT clone?) and there was minimal onward checking of relevance in other breast cancer cell lines – so the general importance remains unknown; 2) the fold change that was used as cut-off (as I understand it) was 1.2 fold – this is low and may reveal false positives; 3) the validation work on SGK1 is interesting but is superficial and needs more work; 4) the siRNA validation work, in particular, needs further specificity checks and controls in some places; 4) the inhibitors used are selective at best, and not specific – therefore further work is required to validate conclusions using these; and 5) I think some things are over-claimed in this manuscript, eg. there are other groups now doing proteomics in combination with on-ward screens (usually CRISPR now).

Specific points regarding the data:

1. Figure 1. Main point here (and true also for some other figures) is that the Legend is not very detailed or informative. This is not an inducible system, and so it is not possible to discriminate between causal signaling events and secondary changes of transformation. This could give rise to drift in culture and at least should be noted in the text.
2. Figure 2. Legend here says information is provided in the Legend! What legend? An informative legend needs to be provided (for all figures). 'Up' and 'Down' is not very quantitative – is there a way of depicting fold change ranges so that the reader can see at a glance the robustness/extent of the changes? Since some of these may only be 1.2 fold – this is important.
3. Figure 3. siRNA screens are notoriously difficult to ensure specificity with many false positives that do not subsequently validate. Did the authors make any attempt to show that hits were as a result of true knock down and what controls were included.
4. Figure 4. I am not sure the 2D viability screens provides much value to the main point of the manuscript and it should perhaps be included only as Supplementary data. At the very least for c, d and e, we would need to see the extent of protein knockdown in both cell types with and without EGF, especially as the authors think that knockdown is different depending on Src and EGF status.
5. Figure 5. What does acinar size actually measure? Is it proliferation, change in cell shape or size, or other features. I think this should be presented and commented upon – it is rather bland to just call it 'transformation', and multiple feature changes may contribute. This figure is weak – especially

there is no quantification, no replicates, of the images in e and the degree of knockdown is not shown, or mention of how the specificity of the siRNA effects is evaluated. This experiment is not adequately described in the text (page 8) and there is no mention of SGK1 in the text in relation to e.

6. The data showing MAP4K5 as a putative suppressor of Src transformation and SGK1 as a putative mediator is the most interesting part of the work presented. Yet the data is somewhat superficial. In Figure 6, b, need to see the degree of overexpression, and it says in legend that 3 siRNAs were used to knock down YAP – yet only one is shown I think. The YAP data is potentially interesting, but be good to know what MAP4K5 and YAP are altering about the acinae –is it proliferation, survival, shape etc. The data in e and f are weak and do not significantly add to the manuscript; it is hard to know what small changes in p-ERK contribute and what the phospho-site changes in f tell us.

7. Figure 7. This is the most interesting data in the manuscript in my opinion, ie. a new role for SGK1 in Src transformation. It needs further work though, eg. what are the multiple bands in top panel in b (why do these bands look different in b and c), the relative changes in SGK1 expression when active Src is expressed is not quantified – the effects of the MEK inhibitor are small and hard to know what this means, d) only one YAP siRNA is shown, what is the degree of overexpression in e (gels are severely cropped), and the SGK1 inhibitor used at high concentrations – what is the specificity at 10-20 μ M? I don't believe this single experiment provides compelling evidence that the changes observed are SGK1 kinase-dependent (as stated in the text on page 10). Better, and more comprehensive, data would help the conclusions, which I think could be interesting – especially if confirmed in other breast cancer cell lines.

8. Figure 8. There is an attempt to examine levels of SGK1 protein in multiple cell lines here with an interesting correlate with cancer sub-type – but it is not related to Src expression/activity and only one Src inhibitor is used to show effects in 2 cell lines. The authors should test all cell lines that express SGK1 with at least 2 Src inhibitors at doses that are unlikely to have off-target effects (we need to see effective Src inhibition). I am not convinced by the conclusion in f that inhibiting both SGK1 and Src will be beneficial generally. This requires further work combining both by genetic knockdown or CRISPR and by combinations of different doses of inhibitors to show synergy (or additive) effects. The one experiment shown is superficial and leads to a rather weak conclusion. This should be expanded, and, if possible, an in vivo (animal) experiment performed.

In conclusion, there is some interesting data in this manuscript, but it should be focussed and it requires more careful experimentation and major revisions before it can be published in Nature Communications. In particular, conclusions about the role of potential phosphorylation changes were weak and should perhaps be dropped.

Reviewer #1 (Remarks to the Author)

This reviewer indicated that ‘Generally, this study technically sounds and contains interesting findings such as the identifications of SGK1 as an important downstream factor of Src and MAP4K5 as a suppressor of Src transformation’ but raised some points that should be addressed.

Major points:

1) For MS-based kinome analysis, the authors used MCF10 A and Src-transformed MCF10 A. However, since Src gene is not necessarily a driver proto-oncogene (there is no significant Y527F mutation in any cancers), this model system may not be suitable for functional annotation of the Src downstream kinases in actual cancer. The authors should use some cancer cells that have substantial upregulation of Src protein, such as A549 lung cancer cells. By comparing these cancer cells with those in which Src function is attenuated by shRNA or some specific inhibitors, the MS-based analysis should be performed. An inducible system may also be useful for shRNA analysis. Furthermore, the authors should add more appropriate explanation for the advantage of MCF10A-Src system. If they need to stick to this system, they should employ inducible systems for WT and Y527F Src to address the role of Src signaling at differential stages of cancer progression.

We accept that Src mutations equivalent to Y527F have not been detected at significant levels at human cancers. However, this mutant has been widely used to identify downstream effects of activated Src (Frame, J Cell Sci 117, 989, 2004), with findings subsequently validated via manipulation of the endogenous kinase. For example, we determined that MCF-10A cells expressing Src Y527F exhibited a similar tyrosine phosphorylation profile to triple negative breast cancer (TNBC) cells, which exhibit a prominent Src family kinase (SFK) signalling network (Hochgrafe *et al*, Cancer Res 70, 9391, 2010). Moreover, many proteins that were tyrosine phosphorylated in the MCF-10A/Src Y527F model exhibited reduced tyrosine phosphorylation in TNBC cells upon inhibition of endogenous SFKs. These data strongly support use of the MCF-10A/Src Y527F model, and additional text explaining this has been added on Page 6, Lines 109-113 and Page 9, Lines 171-176.

To further address this potential issue, as suggested by this reviewer, we have undertaken small-scale kinome profiling of MDA-MB-231 TNBC cells treated with the selective Src inhibitor AZD0530. We would like to highlight that there are potential problems with this validation approach, which include the marked difference in genetic background between these cells and MCF-10As, and contrasting culture conditions. However, while we did not attain the degree of coverage that characterized the original profiling of MCF-10A_Src cells, for sites that overlapped between the two datasets and exhibited enhanced phosphorylation in MCF-10A_Src cells, the phosphorylation of > 70% of these were decreased by AZD0530 treatment of MDA-MB-231s ie the sites were regulated by Src family kinases and in the same direction (Page 9, Lines 176-188, Supp Table 4). In addition to sites on known Src substrates such as FAK, these included sites on TEC, EPHA2, MARK2 and NEK1. These data provide strong support for the validity of the MCF-10A_Src model and our approach.

2) The authors identified MAP4K5 and SGK1 as representative downstream effectors of Src transformation. However, the role of MAP4K5 remains unclear. Although overexpression and siRNA knockdown induced tumor suppressive effects in Src-transformed MCF10A cells, the effects of overexpression of phosphomimetic mutant are limited. Is MAP4K5 really

phosphorylated at S335 in cells? If so, what kinds of kinases phosphorylate MAP4K5? What is the stoichiometry of the S335 phosphorylation? To demonstrate the important role of MAP4K5, more detailed analysis is required. If it is difficult to do so, the part of MAP4K5 should be deleted to make the story more simple.

We have decided that the section on MAP4K5 signalling, while interesting, does not justify inclusion in the revised manuscript. This is because the mechanism of how MAP4K5 affects YAP activity is unclear, and as indicated by all reviewers, the data regarding the MAP4K5 phosphomimetic mutant require further validation. Since we have added a large volume of additional work on SGK1, we have decided to limit the MAP4K5 functional data to the initial identification of it as a suppressor of Src-induced transformation (Fig 4b) and the association of low MAP4K5 expression with poor prognosis in TNBC (Supp Fig 12).

3) MS data show that phosphorylation of SGK1 at S401 is increased only 1.23-fold in Src transformation. The phosphorylation at this site should be verified by other techniques, such as Western blotting with a specific antibody against phospho-S401. Also, Src-induced upregulation of SGK1 protein is no more than 1.5-fold. To support the potential contribution of SGK1 to Src transformation, effects of upregulation of SGK1 on acini formation in control MCF10A cells need to be examined. If SGK1 were a critical driver, it could transform MCF10A cells in a manner like Src.

Phosphorylation of SGK1 S401 promotes, but is not essential for, SGK1 activity (Chen *et al*, J Biol Chem 284, 3453, 2009). In the revised manuscript, we applied a more stringent cut-off to the kinomic data of ≥ 1.5 fold (see response to Reviewer 2), so that phosphorylation of this site is no longer highlighted.

With regard to the activity of SGK1 alone, we have tested this, and SGK1 overexpression in MCF-10A cells does not lead to transformed acini. However, this does not undermine the important biological role of SGK1 in Src-induced transformation. In this context, SGK1 expression is actually required (Fig 4c), and an approximately 2-fold increase in SGK1 expression in MCF-10A_Src cells results in an increase in acinar size of similar magnitude (Fig 5f). We also provide extensive additional validation of the role of SGK1 in Figs 5-9, discussed in more detail in responses to other reviewers.

4) Fig. 8f. To propose the effectiveness of combination therapies using inhibitors of Src and SGK1, the effects of combined treatment with AZD and SGK1 should be examined instead of the combination of SGK1 knockdown and AZD0530.

In the revised manuscript, we provide these data for effects on colony formation by TNBC cells *in vitro* (Fig 7i), and demonstrate a modest synergistic effect. This work is also undertaken with a different selective Src inhibitor (PP2) (Supp Fig 9d), and also on lung cancer cells (Fig 8e), with similar results.

5) Suppl. Fig.2. could be made more easy to read.

This figure has been improved by alteration of the data cut-off (see above) and lay-out.

Reviewer #2 (Remarks to the Author)

This reviewer described the work as ‘interesting’ but raised a series of issues:

the kinome enrichment was performed 3 times but it remains entirely unclear how the authors arrive at their chosen cut-off values above/below which they consider kinase to change.

the chosen cut-offs are tiny, asking the question if such small differences can be robustly detected by SILAC labeling and a triplicate analysis. At the very least, one would expect the cut-offs to be different for different kinase expression/purification levels.

We have revised our approach for determining an appropriate cut-off. This was determined using the same approach as described previously (Croucher *et al*, Cancer Res 73, 1969, 2013), but more stringent criteria applied. The frequency distribution of the phosphosite fold changes was plotted and the data divided into 10 terciles with different percentage cut-offs. A fold change of 1.5-fold was selected as phosphosites defined by this are within the top and bottom 10% of all phosphosites (10th and 90th percentile), respectively. This approach is described in the revised Methods (Page 20, Lines 454-459) and since it limits our list of regulated kinases compared to the original dataset, it has obviously altered the appearance of overview data in Figs 1 and 2.

a worrying statement is that the number of p-sites that are regulated up or down are very similar. This may well imply that simple technical variation is in the same order.

This is not observed with the new cut-off, the number of upregulated sites is much greater (79.5% versus 20.5%).

in addition, they imply that differential kinome enrichment on the resin that they used implies differential activity of the kinases. This may not at all be justified.

We do not make this claim, which would be unjustified. We characterize the expression level and phosphorylation status of specific kinases through a workflow that utilizes both kinome and TiO₂ enrichment. Phosphorylation on specific sites can provide information relating to activation if the functional role of the sites is known.

the authors claim that their approach is novel but it is not. They only cite their own work on the CTx cpd but fail to acknowledge several labs that have come up with similar beads before.

We do not claim the kinome enrichment to be novel, rather the integration of this with a functional screen in order to characterize mechanisms of oncogenic signalling in an unbiased manner (Pages 5-6, Lines 91-104). However, we have included additional references for kinome enrichment prior to MS (Page 6, Line 99).

page 5: a third of phosphorylation regulation on SP sites can hardly be called SRC-specific

This was never referred to as Src-specific. Indeed we state on Page 7, Line 128-130 ‘Activated Src would be expected to initiate a cascade of phosphorylation events, starting with direct phosphorylation by Src itself, and then those mediated by downstream kinase pathways, often involving serine/threonine kinases.’ It would be expected that many stimuli would converge on the latter pathways, but it is still important to characterize their regulation and functional role in the context of oncogenic Src.

large parts of the manuscript are descriptive and we do not learn how these relate to SRC transformation. This reviewer suggests to shorten the descriptive part and focus on the functional validation

This has been done in the revised version. Figs 5-9 focus on functional validation of SGK1.

Speaking of which: Fig 4a is not very clear. Has this data been corrected for multiple testing? How do the kinases relate to each other under the two conditions? in this visualisation, it is unclear which kinase really show a clear difference.

In the revised manuscript we have used ANOVA with Bonferroni's multiple comparisons test (Supp Fig 3a). A highly significant p value is still obtained.

In addition, an additional panel is provided (Supp Fig 3b) that indicates how the dependency on a given kinase varies between control and Src cells. Detailed information for specific kinases is provided in Supp Table 6.

siRNA data: which of the hits that scored in the siRNA screen also showed regulation in the proteomic experiment? this would seem an important piece of information in order to assess the quality of the initial experiment

The functional screen was not kinome-wide, kinases were included in the functional screen on the basis of their regulation in the kinome profiling experiments, although additional proteins were also included based on previous reports of a role in Src-induced transformation. The combined data for the final 10 validated hits from the 3D functional screen are summarized in Fig 4d. However, it should be noted that there may not be a simple relationship between phosphorylation and activity in the functional screen. For example, phosphorylation may be on sites of negative feedback regulation.

*# Fig 7f: how do the authors know that SGK1 cpd is selective? it is used at 10 and 20 uM which are very high concentrations for kinase inhibitors and at which concentration almost no KI is selective
same thing for AZD0530*

The SGK1 inhibitor is used at 10-20 μ M due to low cell permeability, but exhibits strong selectivity at these concentrations (Castel *et al*, Cancer Cell 30, 1, 2016). In the revised paper, we have confirmed a lack of effect on other kinases including the closely-related Akt in our systems (Supp Fig 6b). In addition, we validate the SGK1 data by SGK1 knockdown with si/sh RNA (Figs 4-9, Supp Figs 9-10). We use AZD0530 at lower concentrations (≤ 1 μ M) and in the revised manuscript validate the data using an independent Src inhibitor (PP2) (Fig 8e, Supp Figs 9-10) as well as Src knockdown (Figs 7d, 8b).

Fig1: the data implies that something is going on on Tyr kinases. The rest is speculation and likely just owing to rather too optimistic cut-offs from the proteomic experiments

Please refer to the response above regarding the new cut-off. Using this more stringent approach, approximately 30% of upregulated, and 77% of downregulated, sites correspond to serine/threonine kinases.

Fig 2 does not add anything in particular to the manuscript

We completely disagree. This provides important information regarding the kinase-associated biological processes and pathways that are affected by oncogenic Src. We have improved this figure by including a colour scale that indicates the degree of up/downregulation.

Fig 3a: can the authors explain why the SRC transformed cells also stop growing as do the control cells (albeit at a lower cell density)? would one not expect cell growth just to be slower but eventually go higher than 25% of the controls?

The Src-transformed cells continue to proliferate at a slow rate in –EGF medium, while the control cells stop growing. This is evident from the graph.

Fig 3c is superfluous. How relevant is it to inspect cells after 7 days or 11 day?

This figure has been removed from the revised manuscript.

Fig 4a: we dont learn anything from this figure. it would be important to connect the individual kinases between cotrol and SRC transformation.

We agree and as indicated above have provided this information in the revised manuscript (Supp Fig 3a-b). Detailed information regarding each kinase is provided in Supp Table 6.

Figure 4b uses arbitrary cuts which leads to overinterpretation of the data (essential for both etc)

We agree that these cut-offs are arbitrary but they do highlight contrasting effects of specific kinase knockdowns in the control and Src cells. For example, the observation that Src confers resistance to EGFR knockdown is relevant to the effect of Src on sensitivity to anti-EGFR therapy (Murikami *et al*, Oncotarget 8, 70736, 2017). Different presentations of these data are provided in Fig 3c-d and Supp Fig 3c.

Fig 5b: Given the huge variation in the data, it is unclear if the statistical test actually means anything. Same in Fig 5d. Statistical significance may simply be reached because there are so many data points.

It is well-established that there is variation in acinar size in 3D culture. Consequently, a large number of acini need to be quantified in order to detect a shift in acinar size over the population. The style of data presentation and method of statistical analysis is entirely appropriate. For similar examples in the literature please refer to Clocchiatti *et al*, J Cell Sci 128, 3961, 2015; Anczukow *et al*, Nat Struct Mol Biol 19, 220, 2012; Kurup *et al*, Sci Rep 5, 15153, 2015).

Fig. 6f is not very convincing. why should the Ala mutant have no effect but the phosphomimetic a clear effect? The Ala mutant should show the opposite effect.

We agree that considering the effect of the phosphomimetic substitution, that of the Ala mutant is unexpected. We have not been able to resolve this issue and consequently have removed these data from the revised manuscript.

Fig 7f/g: how can the authors assume their SGK1 inhibitor is selective at the very high

concentrations applied? More controls would be required to establish a role of this kinase in this context.

This point has been addressed above.

Reviewer #3 (Remarks to the Author)

This reviewer indicated that the work ‘has merit and it does reveal potentially novel effectors of Src-mediated event in cancer’.

Specific points raised:

1) the discovery work was done in only one cell line (was it only one clone of Src-transformed MCF-10A cells and one WT clone?) and there was minimal onward checking of relevance in other breast cancer cell lines – so the general importance remains unknown

The kinome screen was undertaken in stable pools of control and Src-transformed MCF-10As. This avoids potential problems in using individual clonal lines. We apologize for not making this clearer in the original manuscript and this detail has been added to the revised version (Page 18, Lines 409-415). In the revised manuscript, we also provide new kinome profiling data derived from an experiment where MDA-MB-231 TNBC cells were treated with the selective Src inhibitor AZD0530. For sites that overlapped between the two datasets and exhibited enhanced phosphorylation in MCF-10A_Src cells, the phosphorylation of > 70% of these were decreased by AZD0530 treatment of MDA-MB-231s ie the sites were regulated by Src family kinases and in the same direction (Page 9, Lines 176-188, Supp Table 4). In addition to sites on known Src substrates such as FAK, these included sites on TEC, EPHA2, MARK2 and NEK1. These data provide strong support for the validity of the MCF-10A_Src model and our approach. See also our response to Reviewer 1.

In addition, we extend the work on Src regulation of SGK1 and the functional characterization of the latter kinase to 2 additional TNBC cell lines and a NSCLC line (Figs 7-8, Supp Figs 8-10), and demonstrate that combined targeting of Src and SGK1 is more effective than monotherapy using a TNBC xenograft model (Fig 9).

2) the fold change that was used as cut-off (as I understand it) was 1.2 fold – this is low and may reveal false positives

As indicated in our response to Reviewer 2, we have now applied a more stringent cut-off. The frequency distribution of the phosphosite fold changes was plotted and the data divided into 10 terciles with different percentage cut-offs. A fold change of 1.5-fold was selected as phosphosites defined by this are within the top and bottom 10% of all phosphosites (10th and 90th percentile), respectively. This approach is described in the revised Methods (Page 20, Lines 454-459) and since it limits our list of regulated kinases compared to the original dataset, it has obviously altered the appearance of overview data in Figs 1 and 2.

3) the validation work on SGK1 is interesting but is superficial and needs more work

As indicated above, we have undertaken a significant body of additional experimental work to extend the characterization of SGK1 to additional TNBC and lung cancer lines, and to determine its functional role in tumour growth *in vivo*. In addition, and as described in more

detail later in this response, we have undertaken additional experimentation to characterize the biological effects of SGK1 knockdown in MCF-10A_Src acini. Immunofluorescent imaging revealed that SGK1 knockdown did not affect proliferation, but it did significantly increase apoptosis within the acini (Fig 6 a-b).

4) the siRNA validation work, in particular, needs further specificity checks and controls in some places

We have provided additional data confirming target knockdown for specific kinases referred to in the text (including EGFR and the final 10 hits from the 3D screen) in Supp Figs 3d and 5, and for individual SGK1 siRNAs from the pool in Fig 4c. In addition, 3 independent YAP siRNAs are utilized to demonstrate the role of YAP in regulating SGK1 (Fig 5d), and we have provided data demonstrating that the effect of inducible SGK1 knockdown can be reproduced with an independent shRNA (Figs 7-8, Supp Figs 9-10).

5) the inhibitors used are selective at best, and not specific – therefore further work is required to validate conclusions using these

As indicated in our response to Reviewer 2, in the revised manuscript we have confirmed a lack of effect of the SGK1 inhibitor (SGKi) on the closely-related Akt in our systems (Supp Fig 6b), consistent with the selectivity of this compound at the doses used (Castel *et al*, Cancer Cell 30, 1, 2016). In addition, we validate the SGKi data by SGK1 knockdown with si/sh RNA (Figs 4-9, Supp Figs 9-10). We use AZD0530 at lower concentrations (≤ 1 μ M) and in the revised manuscript validate the data using an independent Src inhibitor (PP2) (Fig 8e, Supp Figs 9-10) as well as Src knockdown (Figs 7d, 8b).

6) I think some things are over-claimed in this manuscript, eg. there are other groups now doing proteomics in combination with on-ward screens (usually CRISPR now).

To our knowledge this is the first application of such an integrated approach to characterizing the mechanism of action of a specific oncogene, and this is where the novelty lies (indicated on Pages 5-6, Lines 91-104). We appreciate that others may be applying this approach in alternative contexts.

Specific points regarding the data:

1. Figure 1. Main point here (and true also for some other figures) is that the Legend is not very detailed or informative. This is not an inducible system, and so it is not possible to discriminate between causal signaling events and secondary changes of transformation. This could give rise to drift in culture and at least should be noted in the text.

We have improved the quality of the Figure Legends and acknowledge the specific point raised. Additional text has been added to the Discussion (Page 15, Lines 333-337) addressing this point.

2. Figure 2. Legend here says information is provided in the Legend! What legend? An informative legend needs to be provided (for all figures). 'Up' and 'Down' is not very quantitative – is there a way of depicting fold change ranges so that the reader can see at a

glance the robustness/extent of the changes? Since some of these may only be 1.2 fold – this is important.

We have improved the information content of the Figure Legend for this and other figures. Figure 2 has been revised to include a colour scale so that the degree of up and down regulation can be determined. The new more stringent cut-off of ≥ 1.5 fold applies to Figs 1-2.

3. Figure 3. siRNA screens are notoriously difficult to ensure specificity with many false positives that do not subsequently validate. Did the authors make any attempt to show that hits were as a result of true knock down and what controls were included.

Validating all of the primary ‘hits’ from the 2D screen is not feasible, consequently we have provided Western blot or RT-PCR validation of knockdown for specific kinases referred to in the text (including EGFR and all 10 hits from the 3D screen) in Supp Figs 3d and 5, and for individual SGK1 siRNAs from the pool in Fig 4c. In addition, 3 independent YAP siRNAs are utilized to demonstrate the role of YAP in regulating SGK1 (Fig 5d), and we have provided data demonstrating that the effect of inducible SGK1 knockdown can be reproduced with an independent shRNA (Figs 7-8, Supp Figs 9-10). Non-targeting siRNA was always used as a negative control.

4. Figure 4. I am not sure the 2D viability screens provides much value to the main point of the manuscript and it should perhaps be included only as Supplementary data. At the very least for c, d and e, we would need to see the extent of protein knockdown in both cell types with and without EGF, especially as the authors think that knockdown is different depending on Src and EGF status.

We have moved a significant proportion of the 2D screen data to Supp Fig 3 and focused the data presented in the main body of the manuscript on kinases that differentially affect Src and control cells and the contrasting effect of MAP4K5 on the two cell types and the EGF-dependency of this (Fig 3). We have included Western blots confirming knockdown of 5 specific kinases referred to in the text, including MAP4K5 (Supp Fig 3d).

5. Figure 5. What does acinar size actually measure? Is it proliferation, change in cell shape or size, or other features. I think this should be presented and commented upon – it is rather bland to just call it ‘transformation’, and multiple feature changes may contribute. This figure is weak – especially there is no quantification, no replicates, of the images in e and the degree of knockdown is not shown, or mention of how the specificity of the siRNA effects is evaluated. This experiment is not adequately described in the text (page 8) and there is no mention of SGK1 in the text in relation to e.

We have previously described the phenotype of MCF-10A cells expressing active Src in 3D culture. There is morphological variation in the structures formed, but in general the spheroids do not undergo luminal clearance due to a failure of the central matrix-detached cells to undergo apoptosis, and exhibit aberrant distribution of the basal marker Laminin V, as well as the cell-cell junction protein E-cadherin (Bennett *et al*, Oncogene 27, 2693, 2008). These details and the citation have been added to the text (Page 10, Lines 201-204). In the revised manuscript we provide quantification of the data for SGK1 in the validation screen, including a non-targetting control for comparison, and provide Western blot data conforming knockdown (Fig 4c). In addition, we have undertaken additional experimentation to

characterize the biological effects of SGK1 knockdown in MCF-10A_Src acini. Immunofluorescent imaging revealed that SGK1 knockdown did not affect proliferation, as determined by the proportion of Ki67 positive cells. However, it did significantly increase apoptosis within the acini, as determined by imaging for cleaved Caspase-3 (Fig 6a-b), providing an explanation for the decreased acinar size observed.

6. The data showing MAP4K5 as a putative suppressor of Src transformation and SGK1 as a putative mediator is the most interesting part of the work presented. Yet the data is somewhat superficial. In Figure 6, b, need to see the degree of overexpression, and it says in legend that 3 siRNAs were used to knock down YAP – yet only one is shown I think. The YAP data is potentially interesting, but be good to know what MAP4K5 and YAP are altering about the acinae –is it proliferation, survival, shape etc. The data in e and f are weak and do not significantly add to the manuscript; it is hard to know what small changes in p-ERK contribute and what the phospho-site changes in f tell us.

As indicated in the response to Reviewer 1, we have decided to remove the section on MAP4K5 signalling from the revised manuscript. This is because the mechanism of how MAP4K5 affects YAP activity is unclear, and as indicated by all reviewers, the data regarding the MAP4K5 phosphomimetic mutant require further validation. Since we have added a large volume of additional work on SGK1, we have decided to limit the MAP4K5 functional data to the initial identification of it as a suppressor of Src-induced transformation (Fig 4b) and the association of low MAP4K5 expression with poor prognosis in TNBC (Supp Fig 12).

7. Figure 7. This is the most interesting data in the manuscript in my opinion, ie. a new role for SGK1 in Src transformation. It needs further work though, eg. what are the multiple bands in top panel in b (why do these bands look different in b and c), the relative changes in SGK1 expression when active Src is expressed is not quantified – the effects of the MEK inhibitor are small and hard to know what this means, d) only one YAP siRNA is shown, what is the degree of overexpression in e (gels are severely cropped), and the SGK1 inhibitor used at high concentrations – what is the specificity at 10-20 μ M? I don't believe this single experiment provides compelling evidence that the changes observed are SGK1 kinase-dependent (as stated in the text on page 10). Better, and more comprehensive, data would help the conclusions, which I think could be interesting – especially if confirmed in other breast cancer cell lines.

We agree that the data on SGK1 are particularly interesting and important and have made these data the focus of the functional validation.

The multiple bands in Fig 5b and c in the revised manuscript represent different isoforms generated by use of alternative translation start sites or reflect phosphorylation of SGK1. This is clarified in the text (Pages 11-12, Lines 244-246). These bands originally appeared different due to the use of contrasting percentage gels for the two panels. The samples have now been re-run under the same conditions and the difference is not apparent.

The relative changes in SGK1 and its downstream target pNDRG1 are now quantified (Fig 5b).

The effect of the MEK inhibitor is significant (Fig 5c) and we propose that both MEK/Erk and YAP signalling contribute to SGK1 regulation (Fig 5c-e, Page 12, Lines 248-256).

The effects of 3 YAP siRNAs are now shown (Fig 5d).

The fold expression for SGK1 is now indicated (now Fig 5f).

The SGK1 inhibitor is used at 10-20 μ M due to low cell permeability, but exhibits strong selectivity at these concentrations (Castel *et al*, Cancer Cell 30, 1, 2016). In the revised paper, we have confirmed a lack of effect on the closely-related Akt in our systems (Supp Fig 6b) and validate by si/shRNA knockdowns, as highlighted above.

As indicated below, we extend the validation of the functional role of SGK1 to additional breast and lung cancer cell lines.

8. Figure 8. There is an attempt to examine levels of SGK1 protein in multiple cell lines here with an interesting correlate with cancer sub-type – but it is not related to Src expression/activity and only one Src inhibitor is used to show effects in 2 cell lines. The authors should test all cell lines that express SGK1 with at least 2 Src inhibitors at doses that are unlikely to have off-target effects (we need to see effective Src inhibition). I am not convinced by the conclusion in f that inhibiting both SGK1 and Src will be beneficial generally. This requires further work combining both by genetic knockdown or CRISPR and by combinations of different doses of inhibitors to show synergy (or additive) effects. The one experiment shown is superficial and leads to a rather weak conclusion. This should be expanded, and, if possible, an in vivo (animal) experiment performed.

We have included new data demonstrating that the positive correlation of SGK1 with the TNBC subtype is also reflected by an association with increased activity of Src family kinases (Fig 7a). In addition, we have demonstrated positive regulation of SGK1 by Src in an additional 3 TNBC cell lines (Fig 7b-c), using two independent Src inhibitors (AZD0530 and PP2). Effective inhibition of Src is presented in Supp Fig 8. We have undertaken the combination treatment experiment at different concentrations of AZD0530 and SGK1, allowing identification of a weak synergistic effect using the Chou-Talalay method (Fig 7i).

We have also extended this work to another cancer where Src plays an important role, demonstrating that Src regulates SGK1 in the NSCLC cell line A549, and that combined targeting of Src and SGK1 in these cells is more effective than monotherapy (Fig 8). In the TNBC and NSCLC models we validate data using AZD0530 with PP2, and use two different SGK1 shRNAs with similar results (Figs 7-8, Supp Figs 9-10). Finally, we have demonstrated that combined targeting of Src and SGK1 is significantly more effective in a MDA-MB-231 mouse xenograft model (Fig 9).

In conclusion, there is some interesting data in this manuscript, but it should be focussed and it requires more careful experimentation and major revisions before it can be published in Nature Communications. In particular, conclusions about the role of potential phosphorylation changes were weak and should perhaps be dropped.

As indicated above, we have extensively revised the manuscript, including the removal of the work on MAP4K5 phosphorylation sites, and undertaken a large volume of additional experimentation, including animal work, in order to strengthen the functional characterization of SGK1 and provide more focus.

We hope that you now find this manuscript suitable for publication.

Yours sincerely,

Roger J. Daly (PhD).

Reviewers' comments:

Reviewer #1 (Remarks to the Author):

The authors have responded satisfactorily to my comments. I think the manuscript is now greatly improved, and clearly supports the novel conclusion.

Reviewer #2 (Remarks to the Author):

The authors have made many improvements to their original manuscript which this reviewer acknowledges and welcomes.

At the same time, several major concerns remain.

1. as other reviewers have pointed out, we do not really learn if or to what extent this kinase plays a role in SRC-induced transformation because essentially only one cell line is analysed (yes some experiments were done in another cell line)
2. The cut-offs in the proteomic data have been slightly adjusted but implying that this makes the data tighter (because the effects are now in top or bottom 10% of all effects) does not solve the actual underlying issue. At least, the p-proteome response is now less 50/50 which is a good sign but has little to do with statistical significance
3. The authors base quite a bit of their conclusions on the use of Saracatinib because they believe that it is a selective SRC inhibitor. This is clearly not the case. The drug very potently inhibits not only almost all SRC family kinases, but even more potently RIPK2, ACVR1, BMPR1 and several others. In fact, it has about 20 targets at the 1 uM which is the concentration at which the compound was applied. It is therefore simply not justified to attribute the observed effects to the target the authors have in mind. This is unfortunately a very general concern and, therefore, the effects of kinase drugs in particular cannot be properly interpreted without a very thorough assessment of the target spectrum of the molecule.

Reviewer #3 (Remarks to the Author):

I have read this revision and the extensive rebuttal letter carefully.

I think, on balance, that the authors have gone a long way to refining the work in accordance with the points I raised during first review, and the manuscript is improved. I do not comment on the other reviewers' comments. In my opinion, this work adds value to the literature with regard to potential new effectors downstream of an artificially activated Src kinase using state-of-the-art proteomics and screening approaches in MCF10A cells.

The main limitation remains that just one cell line that may not be very typical or representative of triple negative breast cancer was used, and expression of an unregulated Src protein kinase mutant - and so I think the authors should make it clear that the more general relevance of the effects on new putative effectors generally in triple negative breast cancer is not yet clear. A different cell line was tested in vivo with inhibitors of one candidate. Therefore, I think it is important to temper the discussion appropriately with these limitations noted.

The SGK1 results are interesting and more robust in the new version.

Reviewer #1 (Remarks to the Author):

The authors have responded satisfactorily to my comments. I think the manuscript is now greatly improved, and clearly supports the novel conclusion.

Reviewer #2 (Remarks to the Author):

The authors have made many improvements to their original manuscript which this reviewer acknowledges and welcomes.

At the same time, several major concerns remain.

1. as other reviewers have pointed out, we do not really learn if or to what extent this kinase plays a role in SRC-induced transformation because essentially only one cell line is analysed (yes some experiments were done in another cell line)

As highlighted in the first rebuttal, we demonstrate regulation of SGK1 by Src in 4 cell lines in addition to the MCF-10A system (3 triple negative breast cancer lines and 1 lung cancer line) and confirm the functional role of SGK1 downstream of Src in 2 cell lines (MDA-MB-231 breast cancer and A549 lung cancer cells) in addition to MCF-10As.

We accept that the integrated genomic and functional screen were only undertaken in the MCF-10A system and have therefore added the following paragraph to the Discussion (Page 15 highlighted in yellow):

‘In addition, while we have validated the Src-mediated regulation and role of SGK1 in additional cell lines, the global impact of active Src on kinase signalling networks and the dependency of Src-transformed cells on specific kinases may vary according to the genetic background of the cells utilized. We also note that the screen was undertaken with a mutant version of Src not characteristic of human cancers. Consequently, it will be interesting to extend our integrated screen beyond the MCF-10A_Src system to cell lines derived from particular cancer types where Src is known to play an important role in disease progression, and interrogate the role of endogenous, wildtype Src in these contexts.’

2. The cut-offs in the proteomic data have been slightly adjusted but implying that this makes the data tighter (because the effects are now in top or bottom 10% of all effects) does not solve the actual underlying issue. At least, the p-proteome response is now less 50/50 which is a good sign but has little to do with statistical significance

We have provided additional statistical validation in the Methods on Page 20 (highlighted in yellow). This is highly supportive of the stringency of our approach. The text is provided below:

‘A fold change of 1.5-fold was selected as the 135 phosphosites defined by this are within the top and bottom 10% of all phosphosites (10th and 90th percentile), respectively. In addition, statistical analysis of replicates indicated that only 17 sites exhibited a p value > 0.05, and for these, the direction of change for all replicates was consistent (ie an increase for upregulated sites, and a decrease for downregulated sites). In order to avoid false negatives, we have retained these 17 sites within the dataset but highlighted them (Supp Table 1).’

3. The authors base quite a bit of their conclusions on the use of Saracatinib because they believe that it is a selective SRC inhibitor. This is clearly not the case. The drug very potently

inhibits not only almost all SRC family kinases, but even more potently RIPK2, ACVR1, BMPR1 and several others. In fact, it has about 20 targets at the 1 uM which is the concentration at which the compound was applied. It is therefore simply not justified to attribute the observed effects to the target the authors have in mind. This is unfortunately a very general concern and, therefore, the effects of kinase drugs in particular cannot be properly interpreted without a very thorough assessment of the target spectrum of the molecule.

This is a valid point and we have revised the manuscript accordingly on Page 9 (highlighted in yellow). In particular, we no longer refer to AZD0530 as a 'selective' Src inhibitor and we take into account potential off-target effects characterized in the Klaeger *et al* Science paper. The revised text is provided below:

'In order to determine the extent to which our Src-regulated kinome was under Src regulation in TNBC, we undertook small scale kinome profiling on MDA-MB-231 TNBC cells treated with the small molecule Src inhibitor AZD0530. We accept that there are problems with this approach, which include the marked difference in genetic background between the two cell types and contrasting culture conditions. However, while we did not attain the degree of coverage that characterized the original profiling of MCF-10A_Src cells, for sites that overlapped between the two datasets and exhibited enhanced phosphorylation in MCF-10A_Src cells, the phosphorylation of > 80% of these were decreased by AZD0530 treatment of MDA-MB-231s ie the sites were regulated in the same direction (Supp Table 4). While the regulated kinases included EPHA2, LYN, ABL2 and FYN, which are also bound by AZD0530 and hence may reflect off-target effects (Klaeger et al Science 358, 2017), they also included kinases not known to be targeted by AZD0530 and identified by our profiling of MCF-10A_Src cells, including TEC, MARK2 and NEK1. These data support the validity of the MCF-10A_Src model and our approach.'

In addition, and as highlighted in the first revision, we validated the effect of AZD0530 on SGK1 expression, and in combination with SGK1 knockdown/inhibition, using Src knockdown or the alternative Src inhibitor PP2.

Reviewer #3 (Remarks to the Author):

I have read this revision and the extensive rebuttal letter carefully.

I think, on balance, that the authors have gone a long way to refining the work in accordance with the points I raised during first review, and the manuscript is improved. I do not comment on the other reviewers' comments. In my opinion, this work adds value to the literature with regard to potential new effectors downstream of an artificially activated Src kinase using state-of-the-art proteomics and screening approaches in MCF10A cells.

The main limitation remains that just one cell line that may not be very typical or representative of triple negative breast cancer was used, and expression of an unregulated Src protein kinase mutant - and so I think the authors should make it clear that the more general relevance of the effects on new putative effectors generally in triple negative breast cancer is not yet clear. A different cell line was tested in vivo with inhibitors of one candidate. Therefore, I think it is important to temper the discussion appropriately with these limitations noted.

The SGK1 results are interesting and more robust in the new version.

This reviewer is strongly supportive and his/her comments regarding the limitation of using one cell line with a mutant version of Src are addressed in the response to Reviewer 2 (Point 1).

In summary, we have comprehensively addressed both sets of reviewers' comments. We hope that you now find our manuscript suitable for publication,

Yours sincerely,

Roger J. Daly (PhD).